# Cybersecurity in Internet of Medical Vehicles: State-of-the-Art Analysis, Research Challenges and Future Perspectives

**DOI:** 10.3390/s23198107

**Published:** 2023-09-27

**Authors:** Chidambar Rao Bhukya, Prabhat Thakur, Bhavesh Raju Mudhivarthi, Ghanshyam Singh

**Affiliations:** 1Symbiosis Institute of Technology, Symbiosis International Deemed University (SIDU), Pune 412115, India; chidambar.raobhukya.phd2022@sitpune.edu.in (C.R.B.); bhavesh.mudhivarthi.mtech2021@sitpune.edu.in (B.R.M.); ghanshyams@uj.ac.za (G.S.); 2Department of Electrical and Electronic Engineering Science, University of Johannesburg, Johannesburg 2006, South Africa

**Keywords:** Internet of Medical Vehicles, connected medical vehicles, Internet of Vehicles, Internet of Medical Things, connected healthcare, smart cities, cybersecurity, cybersecurity risk in IOT

## Abstract

The “Internet-of-Medical-Vehicles (IOMV)” is one of the special applications of the Internet of Things resulting from combining connected healthcare and connected vehicles. As the IOMV communicates with a variety of networks along its travel path, it incurs various security risks due to sophisticated cyber-attacks. This can endanger the onboard patient’s life. So, it is critical to understand subjects related to “cybersecurity” in the IOMV to develop robust cybersecurity measures. In this paper, the goal is to evaluate recent trends and state-of-the-art publications, gaps, and future outlooks related to this research area. With this aim, a variety of publications between 2016 and 2023 from “Web-of-Science” and “Scopus” databases were analysed. Our analysis revealed that the IOMV is a niche and unexplored research area with few defined standards and frameworks, and there is a great need to implement robust cybersecurity measures. This paper will help researchers to gain a comprehensive idea of this niche research topic, as it presents an analysis of top journals and highly cited papers, their challenges and limitations, the system model and architecture of the IOMV, related applicable standards, potential cyber-attacks, factors causing cybersecurity risks, various artificial intelligence techniques for developing potential countermeasures, the assessment and parameterisation of cybersecurity risks, constraints and challenges, and future outlooks for implementing cybersecurity measures in the IOMV.

## 1. Introduction

Automobile manufacturers consistently try to adopt the latest technological advancements in designing and enhancing automotive vehicles to provide customers with a greater variety of advanced features [1]. Autonomous vehicles (AVs) [2,3,4,5], Intelligent Transport Systems (ITSs) of Vehicles [4,6,7,8], and connected vehicles (CVs) [7,8,9,10] are a few of the advanced features that use new-generation technologies like the Internet of Things (IOT) [11], in which various “devices” in vehicles, like embedded sensors and output devices that use different types of software and technologies, possess processing capabilities to enable communication for connecting and exchanging data with internal and external devices located across the world using the Internet or other communication networks [11,12]. Such vehicles developed based on the application of IOT technology for interconnections are called the “Internet of Vehicle Things” or “Internet of Vehicles (IOV)” [12,13]. By using IOT technology, different devices, applications, and systems that are integrated into the IOV enable V2X (Vehicle-To-Everything) [3] communication with the help of Internet Networks, Wireless Technology, “LAN (Local-Area-Network)”, and In-Vehicle Communication Networks (IVNs) like “CAN (Controller-Area-Network)”, “LIN (Local-Interconnect-Network)”, “Ethernet”, etc., for sharing vehicle-related data with both internal and external devices to develop advanced vehicle features, like vehicle mobility, performance, comfort, efficiency, emissions, and infotainment, and address road safety and security issues [7,8,14]. Similarly, IOT technology is also applied to provide connected healthcare services [15], which is called the “Internet of Medical Things (IOMT)” [16,17]. In the IOMT [18], different medical devices are interconnected to enable machine-to-machine (M2M) communications using different communication networks, like the Internet, Wireless Fidelity (WiFi), Bluetooth (BT), Zigbee, radio signals, etc., with remote healthcare Information Technology (IT) systems to provide healthcare services outside conventional hospital settings as part of connected and smart healthcare [19] in developing next-generation smart cities. By using IOMT technology [20], medical personnel can track and monitor the state of health of a patient to deliver connected healthcare services with more a accurate diagnosis, fewer mistakes, and reduced costs [21,22] when the patient is not present in the hospital, when the patient is located in a remote area, or when the patient is being transported to the hospital in an ambulance, also called a medical vehicle (MV) [21]. 

The purpose of this analysis is based on the research questions, like firstly, what are the recent global trends and state of the art regarding scientific journals on cybersecurity in the IOMV, and how many “types of documents” have been published in this research area? Secondly, what is the architecture and system model of the IOMV? What are the relevant standards and guiding frameworks those are applicable to the IOMV system? Thirdly, to understand what are the different types and characteristics of potential cyber-attacks possible in the IOMV system? Fourth, to know various factors influencing the cybersecurity risk in the IOMV and how can we perform cybersecurity risk assessments and parameterisation to measure the strength of the cybersecurity of the IOMV system? Fifth, to analyse what are the different constraints and challenges involved in developing robust cybersecurity measures in the IOMV? And then find what types of different potential solutions which can help for developing countermeasures for cybersecurity risks in the IOMV system, and what are the future directions regarding this research field?

### 1.1. System of IOMV

MVs are one kind of specific utility vehicle used to transport patients from remote locations to hospitals and provide lifesaving healthcare assistance until patients are carried to hospitals [21]. 

MVs are equipped with some critical, essential medical devices that help to monitor and measure patients’ health conditions, like Body Temperature, Blood Pressure, Cardio and ECG, etc. [21,22], to provide this essential health information and life-supporting medical aid until patients are transported to hospitals for further treatment. Due to inadequate infrastructure, most of the time, trained medical staff may not be available onboard medical vehicles while they transport patients from remote locations to hospitals. So, it becomes a very challenging and critical issue from the patient health point of view. Referring to Figure 1, remotely travelling MVs with onboard essential medical devices are interconnected with communication networks like Bluetooth, WiFi, LAN, and V2X and over the Internet [23] using IOT technology to exchange patients’ health information with medical personnel available at hospitals. This concept is called the “Internet of Medical Vehicles (IOMV)”, which is a futuristic, special application of the IOT that is essentially derived from the integration of IOT applications with connected vehicles and connected healthcare, i.e., “Internet of Vehicles” and “Internet of Medical Things (IOMT)” [24], for realising connected healthcare services as part of developing next-generation smart cities.

The system model and architecture of the IOMV are based on the IOT architecture, which is derived from integrating IOT applications into connected vehicles and connected healthcare, i.e., the IOV and IOMT. As shown in Figure 1, it mainly consists of four layers, i.e., the Data Collection Layer, Communication or Network Layer, Middleware or Data Management Layer, and Application or Business Layer. And this communication can be bi-directional to enable real-time communication and applications for imparting real-time treatment.

Data Collection Layer: In the IOMV system, this layer is also known as the Physical Layer. And its key function is to collect data from various onboard medical devices related to patient health within MVs, in addition to other sensor/functional-related information from the IOV side. These medical devices enable D2D (Device-To-Device) communications and share their information with the Onboard Gateway/IVN, a layer called V2D (Vehicle To Device). D2D and V2D communications take place using Bluetooth, Zigbee, WiFi, LAN, etc., protocols. Next, Onboard-Gateway-to-Onboard-Unit (OBU) communication takes place using IVN protocols like CAN, LIN, Ethernet, etc.Communication or Network Layer: In this layer, when the MV is moving or located remotely on its travel path, it may be required to form multiple and different networks external to the MV to share the intra-MV data collected by the OBU by using V2X communications, such as V2P (Vehicle To Person), V2I (Vehicle To Infrastructure), V2N (Vehicle To Network), V2V (Vehicle To Vehicle), V2R (Vehicle To Road-Side Units), etc. This layer may use the Internet, mobile networks, radio signals, etc.Middleware or Data Management Layer: The data collected from the Communication or Network Layer are further shared with the Middleware or Data Management Layer. In this layer, the collected data are stored and aggregated as cloud data or big data for further analysis and computational analysis in the IOT Computing Layer. Preprocessing and knowledge extraction may be performed using analysis and computational techniques like artificial intelligence and machine learning (AIML). Or, as may be required, the raw information from the Network Layer may be directly shared with the Application or Business Layer, i.e., with the hospital for its analysis. Also, this layer may use the Internet, mobile networks, or radio signals for communication.Application or Business Layer: This is the top or final layer as per the hierarchical order of the IOMV architecture. This layer is the Application or Business Layer, i.e., the hospital setting. The preprocessed data received from the Middleware or Data Management Layer, or the raw data received directly from the Communication or Network Layer as per requirements, will be received by the Hospital Servers. Medical staff like doctors and nurses can analyse these data for their further clinical processing to provide connected healthcare to the patient onboard the remotely moving MV. Further, with the uptake of Electronic Medical Records (EMRs), the exponential adoption of IOT devices in connected healthcare services has increased cyberthreats in the healthcare sector [25,26]. Within healthcare, there has been an explosion in the real-time usage of wired and wireless devices in the care of almost every patient—the IOMT, such as computers, medical pumps, ventilators, anaesthetic machines, operating tables, operating robots, infusion pumps, pacing devices, organ support, syringe pumps, implantable medical devices, a plethora of monitoring modalities, etc. All of these devices, once connected to a hospital network with the IOMV system, as shown in Figure 1, allow the collection of a huge amount of data that can aid decision making, monitor and alert staff to unsafe situations, and expedite patient care. But this interconnectivity presents an opportunity for hackers to attack the systems directly to cause erroneous monitoring, alter the settings of any device, and even access the EMR via the IOMV system using V2X communication, which poses a danger to patient safety.

### 1.2. Cybersecurity Risk in IOMV System

Today, the automotive and medical industries are at an important juncture in history. The IOV is offering exciting new-age advanced features. But the IOMV is prone to incur greater challenges regarding data privacy and potential hazards of cyber-attacks. The security of such shared data transmitted over a network will be at higher risk when malicious hackers try to intrude, attack, and manipulate by exploiting its inherent vulnerabilities. And this can endanger the onboard patient’s life. Therefore, cybersecurity in automotive vehicles will act like a shield by providing protection to the components used in them, like electronic systems and software, communication networks, control algorithms, and user and related data, from malicious attacks, manipulation, etc. [27]. As the IOT is an upcoming technology, most IOT devices do not have end-to-end secured communication connections due to the non-availability of and non-adherence to data security protocols and standards. Also, the communication networks used in automotive vehicles, like the IVN using, for example, the CAN or LIN protocol, are inherently vulnerable and can be prone to cyber-attacks [28,29,30]. And this problem becomes more hazardous in the IOMV because of several issues, like network integration, interoperability, connectivity, availability, etc., as the IOMV [31,32] keeps moving across various locations without being located at a fixed point. Therefore, these IOMV systems are more prone to attacks from hackers located across their travel paths since the communication connections vary, with several en route network points, signal towers, gateway devices, etc. [33,34]. In addition to this, there are no defined standard regulations or guidelines for vehicle-interconnected networks or interconnection protocols for medical devices, including standardised medical devices, which can be used in futuristic IOT applications like the IOMV.

### 1.3. Related Applicable Standards and Frameworks 

The present conventional automotive- and medical-field-related safety and cybersecurity standards are imperfect and are not capable of handling new-age issues like the increasing cybersecurity risk. Therefore, this has led automotive manufacturers to follow their own guidelines, like the International Standard Organisation (ISO) ISO 26262 “Road vehicles—Functional safety” [35] standard, which is mainly focused on the aspects of functional safety only. And it does not cover software development or vehicle subsystems or deal with the concept of risk management of cybersecurity issues. And recently, ISO/SAE (Society of Automotive Engineers) released the first global automotive standard, ISO/SAE 21434:2021 “Road Vehicles—Cybersecurity Engineering” [36], for cybersecurity in automotive vehicles. This standard provides a common framework for addressing the cybersecurity risk in the lifecycle of automotive vehicles and their electronic systems. In addition to these standards, automakers also follow other frameworks, like the National Institute of Standards and Technology (NIST), NIST SP-800-30 [37], and standards by ISO/IEC (International Electrotechnical Commission), i.e., ISO/IEC 31010 [38], to conduct cybersecurity risk assessments in the automotive industry. Similarly, the standard ISO14971:2019 [39] for medical devices outlines the application of risk management to medical devices, software as a medical device, and in vitro diagnostic medical devices for identifying hazards, evaluating risks, and implementing risk controls. The risks associated with medical devices include biocompatibility, data and systems security, electricity, moving parts, radiation, and usability. ISO27001 [40] is the standard for dealing with information security risks and selecting appropriate controls to deal with them. And the standard ISO/IEC 80001—“Application of risk management for IT networks incorporating medical devices” [41]—provides a framework for organisations by using a combined approach associated with the safety of using medical devices and systems on IT networks. And there are numerous other individual standards, like the NIST Cybersecurity Framework, like NIST 800-53 Rev 4 [42], the Information Technology Infrastructure Library, ITIL v3.1.24 [43], ISO/IEC-27000 [44], the European Commission (EC)’s EC Directive 95/45/EC [45], etc. These standards are specific to automotive and medical fields at the individual level. However, these standards are not suitable or wholly applicable to the cybersecurity risks associated with the IOMV. Moreover, the medical personnel in hospitals do not routinely include risk management frameworks in the use of networked medical devices, which are basically supervised by IT specialists. Therefore, issues like uncertainties about non-standard regulations and protocols make the IOMV more vulnerable to cyber-attacks [46,47]. This allows attackers to hack into its systems much more easily and to attack, steal sensitive health information, or manipulate the patient’s health data or readings, medical diagnostics, etc., exchanged between the IOMV and the networks, which can endanger the patient’s life. Hence, it is crucial to define countermeasures to avoid or decrease the potential cybersecurity risks in the IOMV [28,48,49,50,51]. As this concept of the IOMV is one of the futuristic applications of the IOT for realising connected healthcare services as part of smart-city development, it is necessary to understand the existing research works available regarding subjects related to cybersecurity, as well as the issues and challenges involved in it. For this purpose, a comprehensive study and analysis of the existing literature must be carried out.

For this state-of-the-art paper, we have performed the following analysis to contribute to the progression of research in this research area. Our major contributions in this paper include:We evaluated global trends relevant to this research field in terms of top publications, publication patterns, types of journals, top authors in this field, top contributing nations and affiliated institutions, top sponsoring agencies, top keyword searches based on the Boolean technique, gaps, future outlooks, etc., based on a survey analysis of 1582 and 1889 published documents between 2016 and 2023 from the Scopus and WOS databases, respectively.We analysed the top journals and highly cited papers based on the databases WOS and Scopus relevant to this research area and present insights highlighting their methodologies, alongside contrasting each paper’s merits, challenges, and limitations.We conducted a statistical study to analyse the effectiveness of the top-ten countries, and the results are discussed.We define and present the system model and architecture of the IOMV system and types of communications and present the relevant applicable standards, protocols, and governing frameworks.We describe a variety of factors that are responsible for causing a cybersecurity risk in the IOT-application-based IOMV. We discuss how to perform a risk assessment and parameterisation to strengthen the cybersecurity of the IOMV system and describe solutions helpful in addressing cybersecurity risks, like using AIML, artificial general intelligence (AGI), etc.We classify and consolidate different types of sophisticated potential cyber-attacks in tabulated form; these attacks can occur at various layers of the IOMV, right from the source, i.e., onboard IOT-connected medical devices, up to the target recipient devices at hospitals and their interconnecting communication channels and networks. And we explain the cause and impact of various significant cyber-attacks.In the last section, we provide the limitations of this work, major constraints, and significant potential challenges that need to be addressed and discuss the outlook for future work related to implementing robust cybersecurity measures in these IOT-application-based vehicles.

Further, this paper is organised as follows: Section 1 describes the introduction, research questions, system model, architecture, cybersecurity risk, and related applicable standards and frameworks of the IOMV system. Section 2 lists articles in the existing literature in this research area. How the study of the existing literature was conducted, using roadmaps and the research methodology, is explained in Section 3. Section 4 demonstrates the scrutiny of the literature review analysis and debate gaps. Section 5 discusses techniques to assess the cybersecurity risk in the IOMV system, the types and characteristics of cyber-attacks, the application of AIML techniques for strengthening cybersecurity in the IOMV, and parameterisation for measuring the strength of cybersecurity. The results and limitations of this research work are discussed in Section 6 and Section 7. Constraints and challenges, along with trends in this area of interest, are discussed in Section 8. Section 9 and Section 10 provide the conclusion, indicating guidelines regarding future implementations. 

## 2. Primary Data

The following research analysis in this article is a “quantitative study” performed based on databases of publishing bodies, viz., Scopus, ABCD, EBSCO, CrossRef, Web of Science (WOS), and PubMed, and different types of documents, like Journal Articles, Conference Papers, Review Papers, Chapters of Books, Books and Short Studies, Conference Reviews, Editorials, etc. The literature analysis helps academics identify shortages in performing research in particular fields [52]. This analysis uses scientific methods, including mathematical representations and analytical tools, to assess the results. The literature study for the analysis in this research article, “Cybersecurity in Internet of Medical Vehicles”, explored one of the futuristic applications of technologies from the Internet of Things; this application is based on the integration of other IOT applications, the IOV and IOMT, to realise connected healthcare services as part of developing smart cities. To conduct this literature analysis, the databases WOS and Scopus were considered. A similar study was performed for different research areas and databases, as mentioned in reference [53]. The empirical data collection of all scientific publications from 2016 to 17 August 2023 was considered for this literature study. The combination of keywords used in the search field is “Internet of Things” AND “Cyber Security” OR “Internet of Vehicles” OR “Medical Devices” for the period between 2016 and 2023. The top-15 keywords reflected in this research study are listed in Table 1 with the number of documents published. The citations used in this database are from a large variety of domain fields, like Biomedical Engineering, Electrical Electronics Engineering, Healthcare Sciences Services, Computer Science Cybernetics, Legal Medicine, Emergency Medicine, Computer Sciences, Engineering, Biochemistry, Chemistry, Astronomy, Energy, Physics, Material Science, Decision Making, Social Sciences, Mathematics, etc. This paper helps research members like professors, manufacturers, scholars, and new and upcoming researchers in this field to gain an overview of the research topic of “cybersecurity in IOMVs”.

A systematic and complete analysis of literature data has given rise to several literature reviews in various areas. Various tools are used to derive relevant, useful data and to make visualisations. Analytical tools are used to find the research gaps and identify correlations among authors and keywords, making a literature analysis network.

Table 2 provides various corresponding analytical methods and databases for studying “Cybersecurity in Internet of Medical Vehicles”. An analytical study of the literature lets a person who reads find key research domain variables quickly. In [54], the authors give the vision, applications, and future challenges of the Internet of Things. But integrating the IOT has many constraints and challenges because there are many new and untapped research areas in this sector. In [55], the authors describe the analysis of a literature study explaining the interaction between AVs and manually driven vehicles, as well as the safe and efficient operation of AVs. The authors illustrate AVs with their challenges in sustainable urban mobility, like security vulnerabilities and privacy [56]. In [57,58], the authors discuss the gaps and opportunities in healthcare and cybersecurity. In [59,60,61], the authors propose blockchain technology (BCT) as one of the solutions for overcoming the challenges and constraints in implementing the IOT and Internet of Medical Things. Upon knowing this topic, several issues in blockchain problems can be resolved. In [62,63], the authors analyse various trends in cybersecurity. In [61], the author explains the implementation of smart healthcare services using the Internet of Things (IOT). Moreover, in [46,64], the authors present the architecture and communication models of the IOT, focusing on different types of “security attacks” at different layers of “the IOT system” and “countermeasures” that can be taken, whereas in [65], the authors describe security issues in developing smart automobiles, their attacks, and solutions and categorise different types of cyber-attacks and security threats in Vehicle Ad hoc Networks (VANETS) and the IOT [66].

In [67,68], the authors explain V2X communication and the integration of sensor data from vehicles for AVs using “localised vehicular-communications”, “cloud-connectivity,” and “long-range-cellular-networks”. In [20], the authors identify and compare the existing “Healthcare IOT (HIOT)” systems and classify them into five categories, viz., “resource-based”, “sensor-based”, “communication-based”, “security-based”, and “application-based” approaches. Also, they evaluate various merits and demerits of the selected methods and contrast them in evaluating techniques, tools, and metrics. At the same time, in [16,69], the authors discuss combining the IOT and telemedicine for implementing smart healthcare. And they explain the concepts of the IOT, its architecture, different network layers, their processing, and communication systems in remote and connected healthcare services.

In [70], the authors identify and classify different “attack-mechanisms” regarding various electronic components used in AVs, such as potentially affected domains in them, along with the profiles of the attackers. Also, they address information security principles, which help to address attack-mitigation steps. However, this study is confined to the security risk for the IOV only. In [71], the authors analyse the penetration of mobile health in the Indian context, such as health services using telemedicine, mobile phones, etc., and identify the low adoption of mobile health due to very little awareness of the word m-Health. And this analysis is based on a single nation, i.e., a perspective from India’s point of view only, not a global-level analysis. In [72], the authors emphasise the role of big data in the IOMT, i.e., healthcare with the IOT, related to patients’ health data, involving diagnosis, maintenance, and management. But this analysis emphasises data generated from interconnected medical devices from hospitals alone. The authors mention the usage of 5G technology for providing smarter healthcare and facilitating better quality, monitoring, and tracking in a more efficient and sustainable manner in applications such as “telemedicine”, “telesurgery”, “robotic surgery”, “connected ambulance”, etc., but this review mentions the 5G network application perspective in healthcare. In [73], the authors analyse the properties, evolution, and trends in autonomous vehicles and identify gaps and provide suggestions for improvements in terms of policies and regulations, cybersecurity related to the risk of data privacy and security breaches, attacks, etc. The authors analyse developments in the application of Information Technology (IT) in the healthcare system from 2004 to 2022 and emphasise the need for robust cybersecurity systems [58]. In [74], the authors explain that the IOMT is an application of the IOT in the healthcare sector in which data are transmitted across various communication networks. However, it is highlighted that this technology is still in the development phase only and yet to become mature, and there are security risks because of poor maintenance, insufficient standards, legacy devices, and equipment fitted in medical centres that are not designed for an Internet connection, coupled with little user awareness. And this can allow hackers to easily attack and control less secure IOMT devices, such as accessing an unencrypted IOMT device through unencrypted communication. And their research results highlight that artificial intelligence and machine-learning techniques, such as blockchain technology, can effectively address the security concerns of the IOMT [75]. But this study is limited to the security risks in the application of the IOT for medical devices connected to hospital networks only. 

Table 3 provides an analysis of various review journals related to the research field of “Cybersecurity in IOMVs”, which illustrates their novelties, contrasts relevant advantages and disadvantages, and highlights their research gaps. In addition to these, in [32,76,77,78,79], the authors explain the development of the Internet of Vehicles for autonomous-vehicle applications for achieving a safer, better, efficient, and transformative future “Intelligent Transportation Systems (ITS)” for the realisation of smart mobility and smart cities [19]. These vehicles are equipped with sensors like radar, cameras, etc., and communicate their data with other vehicles and infrastructure based on the IOT using V2X communication technology, which can be an unreliable wireless and mobile network. In the IOV, communication occurs among onboard computers called “Electronic Control Units (ECUs)”, and their corresponding sensors and actuators form an In-Vehicle Network. Owing to their apparent simplicity, low cost, and reliability, CAN and LIN protocols are extensively used as default standards for IVN communication [80]. Failures and attacks in the networks of connected vehicles could also affect their connected things, and they can also become vulnerable and prone to various security attacks: e.g., an attempt to misguide inputs can cause them to be exchanged with other nearby connected vehicles, which can lead to hazards [27]. Security researchers specify that cyber-attacks can be broadly made at two layers, i.e., “in-vehicle threats” regarding the internal devices of the vehicles and “communication-threats” that the connected vehicles in the network of the IOV use to communicate for the proper functioning of their subsystems: e.g., an attacker can manipulate engine control by tampering with the vehicle IVN, which can potentially become harmful, as the receiving node does not validate the origin of a CAN message [80].

During the period of 2021–2023, many different highly cited journals published articles in a wide variety of domains, viz., global geoparks analysis [82], eye health/vision impairment [83], policy analysis using data science [84], the consequence of knowledge management in Industry 4.0 [85], security aspects corresponding to the smart grid [86], transfer pricing [87], etc. The study and analysis of existing literature works provide an overview, and they give only shallow insights into any article. On the other hand, a review of a journal paper gives a deeper understanding. In [88], the authors detail how a literature study analysis is performed. 

## 3. Research Methodology

The literature analysis in this paper is organised into 5 phases: the objective and design of the study, the collection of data, data analysis, the visualisation of the data through various tools, and the interpretation and comprehension of the data, as presented in Figure 2. A literature review analysis is a scientific and systematic procedure implemented using mathematical, computational, and statistical analyses. The foremost objective and purpose of conducting this research is to explore which authors, nations, journals, keywords, years, subject areas, and leading languages and institutions are dominant in the fields of IOT, IOV, IOMT, IOMV, connected healthcare, smart cities, data privacy, and cybersecurity issues involved in integrating them.

The initial stage began by raising research-based questions. The selection of relevant and applicable keywords for analysis was also a major issue. The relevant keywords were finalised by referring to the topmost cited publications with critical keywords as per the authors [74]. Applicable and related databases from WOS and Scopus were searched and browsed for review journal articles associated with IOT, IOV, IOMT, connected healthcare, smart cities, data privacy, and cybersecurity issues involved in integrating them.

In the second stage, the appropriate, relevant keywords were collected by filtering key points, like the publication type, subject, year, keyword, citations gained, author, and affiliated body. The Boolean technique was used for searching keywords to obtain varied outcomes for the documents. For example, entering an “inverted comma (“ ”)” in each keyword search does not let two or more words in a block of words separate, and likewise, if a “single colon (‘’)” is used, it resembles a “suffix” along with a keyword for the search to obtain the desired results. We set “year constraints” as selection criteria for data mining, with the investigation’s period constrained from 2016 to 2023.

In the third stage, the analysis of the extracted data was performed. The process of normalisation of the data was performed to check and correct data that were irrelevant/missing/oversampled/redundant from the downloaded CSV file; then, the data network was mapped before proceeding to further steps. 

In the next phase, i.e., phase 4, the visualisation of the data was performed. In the final step, interpretations were made, and inferences were derived based on available visualisation results. The derived outcomes help us to make conclusions regarding research-based “questions & answers”. Also, inference and comprehension from the literature study provide insights to identify gaps in the literature, possible trends in the future, and in-depth details of the work that has already been conducted in the chosen particular areas of research. This literature study and analysis accounts for “scientific & systematic” publications in a specific subject area within a shorter period. 

### 3.1. Important Keywords

Important “keywords” used in the search are “Internet of Things” AND “Cyber Security” OR “Internet of Vehicles” OR “IOT in Vehicles” OR “Medical Devices” AND “IOT in Vehicles” OR “IOT in Medical Things”, OR “Internet of Medical Things” OR “Security in IOT” OR “Cybersecurity in Medical Things”. “Primary and secondary” keywords used are mentioned in Table 4. When using only “primary keywords”, the search resulted in 1593 documents in the DB “Scopus” and 1889 documents in the DB “WOS”. Then, using specialised keywords for a particular topic, i.e., combining primary and secondary keywords by “ORing”, resulted in 2021 documents in “Scopus” and 206 in “WOS”. Employing the above “query-sting” keywords with the “Boolean technique” side by side with the years limited between 2016 and 2023, a total of 1203 documents were available in “Scopus”, and 30 documents were found in “WOS”.

On the Scopus and WOS websites, options are available to consider or discard or restrict country, language, affiliation, types of publication, and several filtering options. The precise amalgamation of “keywords” simplifies classifying the target research areas in terms of the “Number of Publications (NOP)”.

The resulting documents show that the “WOS” DB has a greater number of documents than the “Scopus” DB. The pursuit of relevant and fitting keywords plus the time range for which a “keyword” is searched, moreover, plays a substantial role in identifying appropriate data. “Tips” for searching “results” in diversified ways are described in Table 5, which aids researchers in refining and zeroing in on their research topic.

### 3.2. Preliminary Analysis

Figure 3 shows documents published per annum from 2016 to 2023. The database WOS has more publications than the Scopus database. There were steep increments in the NOP in both databases, i.e., from 2016 to 2022, and most publications took place in the year 2022, with 404 compared to approximately only 33 in 2016 in the Scopus DB and 463 in 2022 compared to only 216 in 2016 in the WOS DB. And the data within the second quarter of 2023, with around 89 publications in the DB “Scopus” and 48 in “WOS”, indicate that the number of documents published every year is growing, which demonstrates that this area of research is an outstanding field for steering future potential research work.

Figure 4 presents the top-10 most influential authors in “Cybersecurity in Internet of Medical Vehicles” from both Scopus and WOS databases. Ross, J.S., is the author with the top number of publications, with a total of 27 published documents, trailed by Krumholz, H.M., with 17 publications in the WOS database. Javed A. R. is the most contributing author in the database Scopus with 19 publications, followed by many contributions by the authors Jalil, Z., Maple, C, and Zulkermine, M, with 14 publications. And the author Guizani, M., is the common significant contributor in both Scopus and WOS databases, with 11 and 10 publications, respectively. Figure 5 presents the top-10 affiliations in the DBs “Scopus” and “WOS”. The University of Texas at San Antonio is the front-runner with 25 publications, trailed by Beihang University, Coventry University, and the Air University of Islamabad with 22 publications in the Scopus DB list. Harvard University leads with 42 publications, followed by the University of California System with 38 and Yale University with 37 in the WOS DB list. Figure 6 and Figure 7 present the categories of documents with the number of “types of publications” in the DBs “Scopus” and “WOS”, respectively. In the Scopus DB, articles have more significant numbers, with 776 publications at around 51.7%, followed by Conference Papers and 481 journals at about 32.0%, while in the WOS DB, articles also have the largest number of publications, with 1105 published documents at around 54.12%, followed by Proceeding Papers with 482 published documents at 25.8%, Review Articles with 205 published papers at 10.97%, and categories like Editorials 3.96%, Materials, Early Access 2.52%, Book Chapters 2.03%, and others 0.61% etc.

Figure 8 illustrates the top-10 countries with the number of publications. China has the largest NOP with 352 published documents, followed by the USA in second position with 348, the UK in third with 156, and India in fourth place with 154 papers published in the Scopus DB. At the same time, the USA leads with its NOP at 571 in the WOS DB, followed by China with 179, the UK with 139, and India in fourth position with 123 published documents.

According to the analysis of the Scopus DB, Table 6 lists the top-10 nations with the highest number of publications and the most significantly cited papers. The “statistical *t*-test” calculates the “variance” in productivity among the top-10 nations. The “null hypothesis (H_o_)” is that there exists no variance amongst NOPs of top-10 nations with outstanding cited papers or the max NOPs by the top-10 nations. Similarly, the “Alternative Hypothesis (H_a_)” is that there exists a variation amongst NOPs of top-10 nations with significantly cited papers and the max NOPs by top-10 nations.

A one-sample *t*-test was performed, and Formula (1) below shows the t-ratio:(1)$$t=\frac{\left(\bar{X}-\mu \right)}{\left(\frac{s}{\sqrt{n}}\right)}$$

where:

$\bar{X}$ = “calculated mean”;

μ = “hypothetical mean”;

*S* = “standard deviation”; 

*n* = “sample size”.

JASP is open-source software and is freely available for downloading. JASP software was used to perform the statistical analysis for this article, and its results are illustrated in Table 7 and Table 8. The calculated “*p*-value” is greater than 0.05 for both sample publications. This means that the “Null Hypothesis (H_o_)” is accepted, and the “Alternative Hypothesis (H_a_)” is rejected. It shows that the NOPs of the top-10 nations are the same as the highly cited papers and max NOPs by top-10 nations.

Figure 9 and Figure 10 present the top sponsoring agencies from 2016 to 2023 with the number of documents published from the Scopus and WOS databases. The sponsoring agency “The National Natural Science Foundation of China” leads the lists of Scopus and WOS databases with 151 and 77 publications, respectively. They were followed by the “National Science Foundation” with 76 publications in the Scopus DB and by the USA “Department of Health and Human Services” with 72 publications in the DB WOS. Figure 9 and Figure 10 show that most sponsoring is by “NSFC China”. Figure 10 reveals that most of the research papers are published by institutes from China.

Figure 11 presents the NOPs in the DBs Scopus and WOS relating to the subject area. “Engineering” has larger NOPs in both DBs. A total of 1163 documents linked to engineering were published in the DB WOS, followed by 328 “Computer Science” documents. A total of 696 documents were linked to “engineering” in the Scopus DB, followed by “Computer Science” with 1136 documents, whereas Figure 12 and Figure 13 present the proportions of the subject areas based on the databases Scopus and WOS.

## 4. Literature Review Analysis

The “citation index” indicates how many times a paper is referenced by other research members for their research work. Likewise, the “h-index” is one of the indicators that helps to evaluate how “productive” and “influential” authors are. The “h-index” is calculated when the NOP published by an author cited is more than or equal to the NOP published by the author.

Table 9 and Table 10 list the top-15 publications that have the highest number of citations based on the databases Scopus and WOS, respectively, provided side-by-side with “year of publication”, “title”, “author name”, “journal publication”, and “total citations” collected from 2016 to 2023. Stergiou, C., et al., in the Scopus DB, for their review paper “Secure Integration of IOT and Cloud Computing”, were cited the highest number of times, with 739 citations. In contrast, Yang, Y.C., et al., for their survey paper “A Survey on Security and Privacy Issues in Internet-of-Things”, leads the list of the WOS DB, which gathered 572 citations. Both papers discuss security issues in integrating the IOT, which shows that cybersecurity in the IOT is a highly discussed and researched interest topic by research members across the globe.

Many papers are available with similar works, which have made limited contributions to the research field. While comprehending a particular paper, its abstract and conclusion can be read to understand its objective and focus area. If the topic and subject seem helpful, then one can continue to study other details of the paper. While inspecting the complete details of the paper, the reader should first go through an overview of the debated topics in all other segments, figures, and tables to understand the pattern and to identify the relevancy of the work of that particular paper. Through this, the contribution of the paper to the research topic can be understood. Each paper is exceptional. This method must be adhered to while reviewing the literature (LR).

The literature review step provides insights into the qualitative contribution of articles, their outcomes, gaps in research, methods used in papers, the assessment of various procedures, and employed approaches. On the other hand, the literature review step emphasises the quantitative analysis of the papers, their authors, contributing countries, publication years, briefs about a focus area, used keywords, sponsoring agencies, journals, and affiliations with their institutes. 

Figure 14 presents a crucial evaluation of the top-cited paper on “Scopus & WOS” DBs. The authors with more citations discuss a critical analysis of issues, challenges, and methodologies to address cybersecurity in the Internet of Medical Vehicles. Predominantly from 2022, these publications are mostly cited by other research members, which suggests that researchers are very excited about this cybersecurity-in-IOMV topic.

“Security” in the Internet of Things (IOT) has a total of 2021 papers in Scopus and 206 in WOS. Simultaneously, cybersecurity in medical devices has 28 papers in Scopus and 164 in the WOS DB. The difficulty level for addressing security issues for implementing the Internet of Medical Vehicles is higher than that for the IOT alone. But practically, more published documents are available for the latter, and very little research has been carried out regarding security issues and challenges.

Figure 15 and Figure 16 present the graphical trends of the top-5 published documents in the databases “Scopus & WOS”, respectively, in cybersecurity in the IOMV. The paper titled “Secure Integration of IoT and Cloud Computing” in 2019 and “A Survey on Security and Privacy Issues in Internet-of-Things” in 2021 had steep rises in the number of citations from the databases Scopus and WOS, respectively. In Figure 15, it can be observed that the papers titled “Deep Learning for cyber security intrusion detection: Approaches, datasets, and comparative study” and “A Survey of Machine and Deep Learning Methods for Internet of Things (IoT) Security” have gathered a more significant number of citations over the period up to 2022. And likewise, in Figure 16, the number of citations for the papers titled “An effective feature engineering for DNN using Hybrid PCA-GWO for intrusion detection in IoMT Architecture” and “Internet of Medical Things: A Review of Recent Contributions Dealing with Cyber-Physical Systems in Medicine” increased in 2022. This illustrates that the tendency of highly cited papers changes from year to year. In that specific publication year, whichever paper was significantly cited upheld that it had considerably new, valuable substance during that period.

Figure 17 and Figure 18 explain the “Source Titles” from the databases Scopus and WOS, respectively. “IEEE Access” has a greater percentage of contributions with 6.03%, followed by “IEEE Transactions on Intelligent Transportation Systems” with 2.45% in the Scopus DB, whereas, in the database WOS, the “Journal of Medical Devices Transactions of the ASME” with 9.95% publication contributions has the highest percentage of publications, followed by “Expert Review of Medical Devices” with 7.81%.

## 5. Assessment of Cybersecurity in the IOMV System

This section discusses the assessment of cybersecurity in the IOMV system, like details regarding various types of potential cyber-attacks and their characteristics, the application of AIML techniques for developing countermeasures to cyber-attacks, the assessment of cybersecurity risks, and parameterisation for estimating the strength of cybersecurity.

### 5.1. Types and Characteristics of Cyber-Attacks

The application of the Internet of Medical Vehicles is essentially developed based on the application of IOT technology. Table 11 represents various potential sophisticated types of cyber-attacks that can be carried out by malicious hackers and attackers at various layers of the IOMV, right from the source, i.e., onboard medical devices, up to the target recipient devices at hospitals through their interconnecting communication channels and networks, including common attacks that can occur at any layer of the IOMV system [89,90]. A variety of network devices communicating through vehicle IVN and wireless communications (e.g., Gateways, Network Nodes, WiFi Devices, Servers), IOMV devices connected at the physical level (like medical devices, HMI, USBs, etc.), their software and firmware, etc., will be part of the IOMV network, which potentially allows more possibilities for cyber-attacks in the IOMV system, such as denial of service (DOS), Packet Injection, Spoofing Attacks, IVN bus flooding with overwriting messages, etc., leading to security and privacy risks to the end users and the shared data. Because the data that are transmitted are open to all third parties, and the attacks can be performed from a remote distance, either from the roadside (Roadside Units) or from other vehicles in the IOMV network, etc., the resultant consequences of different types of attackers, like internal, rational, external, individual, destructive, random, lasting, opposite, etc., will altogether vary. Hence, it may lead to a rise in newer types of combinational and sophisticated attacks that can exploit vulnerabilities of the IOMV system.

While IOMV systems have greater potential to transform human life, they pose significant security concerns, and they can become vulnerable targets for attackers. Thus, interest in the security of the IOMV has been increasing rapidly. Also, various factors influencing the cybersecurity risk in the IOMV, like the operating conditions, integration, interoperability, restraints, and user necessities for vehicular communication (VC) systems, make security an increasing challenge. Cybersecurity regulations are essential to any Information and Communication Technology infrastructure [77]. 

However, the effectiveness of the countermeasures applicable to the IOMV has not been evaluated yet. So, as per this analysis, there are many issues that are still open that need to be addressed, like protecting IOMV systems from outside attacks and securing V2X communication, as they can spread to the smart infrastructure level and vice versa. Also, the capability of responding in real time, suitability, affordability with minimum changes to the architecture of the IOMV, etc., can be requirements for robust security enhancement strategies, which require new techniques and standards that are not based on outdated transportation protocols and that can work successfully in realistic, challenging, and niche environments like the IOMV system.

There is a big challenge in how to implement this without compromising safety, reliability, security, and privacy. To address these challenges, effective detection and defence mechanisms and countermeasures, like technologies and policies that can reduce the risk of security vulnerabilities at the vehicle level and infrastructure level, must be developed and assessed against various types of attacks by collaborating with government agencies, academicians, and industry. Various stakeholders of the IOMV system, like the manufacturers, are responsible for the development of safety countermeasures applicable to the IOMV system in addition to the development of countermeasures applicable to infrastructure, requiring more effort from governments and regulating agencies as well.

### 5.2. Artificial Intelligence and Machine Learning for Strengthening Cybersecurity in IOMV System

The conventional standard algorithms needed for decision making, which are designed based on hard-wired logic to develop cybersecurity defensive countermeasures for combating quickly growing intelligence and dynamically evolving sophisticated cyber-attacks, are not sufficient alone. And it is not possible to detect all the threats or to handle such cyber-attacks by human efforts alone [91,92]. Such algorithms might be slow and insufficient, owing to the speed of processes and the quantity of information that needs to be processed. Human security experts may not be capable of handling the degree of security threats [92,93]. Therefore, they must have strong support from intelligent machines, software, and the latest technologies for fighting cyberthreats [94,95]. Therefore, cybersecurity capabilities can be enhanced by the usage of artificial intelligence and machine-learning (AIML) techniques. AIML comprehends and simulates human intelligence [92]. AIML techniques can improve the overall security performance, as they are more flexible, adaptable, and robust. And they help to provide better protection from an increasing number of sophisticated cyberthreats [96,97]. It is necessary to advance in AIML applications since the increasing volume and complexity of cyber-attacks require more resources to address issues of cybersecurity. And AIML algorithms can analyse the data and learn to identify threats from existing attacks and can detect intrusions with the same patterns in the future and limit the risk of breaching [98,99]. To combat cybersecurity risks in IOT/IOMV applications, AIML techniques deal with machine learning, deep learning, natural language processing, computational intelligence, etc. [93,100]. 

AIML techniques have the capability to replicate human intelligence. And they are efficient in Automation, Smart Decision Making, Accuracy, Exploration, Data Collection, Data Analysis, Solving Complex Problems, Managing Repetitive Tasks, Minimising Errors, etc. And AIML has emerged as one of the most crucial tactics for addressing issues of cyber-attacks. When deployed for cybersecurity management, they can identify threats and examine relationships among cyberthreats quickly, and they accelerate detection. AIML techniques work efficiently in scanning the entire network very fast, detecting potential threats, and shortening safety tasks, unlike humans. AIML techniques help in eliminating cumbersome tasks and make instant decisions to remediate threats. AIML applications are capable of handling huge amounts of data, and hidden threats can also be screened and handled. They provide enhanced security and vulnerability management. And they perform various tasks, including data loss prevention, threat identification, security management, vulnerability management, fraud detection, access management, threat and compliance management, intrusion detection, threat intelligence, etc. They can address various types of threats and help to prioritise them to safeguard the whole system. AIML helps in securing validation and authentication each time someone performs a login activity for the recognition of genuine login attempts through fingerprint identification, face recognition, OTPs, etc. [92]. Data trends can be identified, which also enables security systems to make informed decisions. And AIML techniques help to reduce response times and enhance security measures in cybersecurity issues [92,101].

The authors have explained that due to the transmission of large-scale data through IOT devices in the IOMV system, AIML plays a primary role in anomaly detection, addresses the types of cyber-attacks, and provides defensive countermeasures. And the applications of deep learning (DL) and deep neural networks (DNNs) are possibly the most promising and powerful tools among AIML techniques [92,95,102,103]. Deep neural networks (DNNs) may be able to predict cyber-attacks in advance, in addition to preventing them. And this can lead to a new phase in addressing the cybersecurity risk in the IOV and IOMT [26,103]. Also, BCT intelligence, which combines the advantages of both BCT and ML techniques, is helpful for the rapid development of IOV security systems. Driven by its advantages, four aspects, namely, reliable interaction, network security and data privacy, trustworthy environment, and scalability, can be achieved by BCT [104]. The authors have summarised potential vehicle-level cyber-attacks in the IOV, both within IVNs and mobile networks. Further, they have presented a centralised third-order model and designed resilient-based adaptive countermeasures against cyber-attacks for controlling the IOV system and mitigating the limited network communication bandwidth under the threat of denial-of-service (DOS) attacks, deception attacks, etc. [105,106].

And regarding IOMT applications, most IOMT devices are small and easy to access, and they suffer from issues like latency and low power consumption, and thus, the IOMT needs new measures to keep the system secure [101]. Blockchain techniques can consistently identify the most severe and potentially life-threatening attacks in the IOMT field. The use of blockchain for decentralised data protection helps to protect patient health records from being compromised. And there is an emphasis on focusing on integrating DL, ML, and BCT techniques to improve cybersecurity by utilising lightweight solutions like a decentralised digital ledger approach that allows end-to-end communication and interaction between untrustworthy persons. This technique uses BCT to record and collect accumulated medical information in a secured and distributed format from the IOMT and integrated devices [107]. Another way to improve cybersecurity is integrating blockchain with currently used safety measures, like authorisation and access control providers [108].

The authors have proposed a comprehensive combination of traditional and Quantum Neural Network/ML (QNN) techniques as part of AIML in the IOMT vulnerability assessment. Specifically, the protection of sensitive and private data at multiple levels can start from storing data in distributed cloud nodes and then fuse heterogeneous IOMT data by using ML classification algorithms such as QNN to predict which vulnerability in IOMT-based network traffic is a malicious attack [109,110]. In [111], the authors highlight types of cybersecurity attacks related to the IOT, especially those using RFID and WSN technologies, which are the basic technologies used in IOT applications. In [104,112], the authors emphasise that BCT and ML can effectively address the current issues of decentralisation, cybersecurity, and data privacy in the IOMV. In [104], the authors propose token-based authorisation and authentication (TAA) for providing cybersecurity in IOV systems. This method uses AIML techniques like BCT and Random Forest learning for authorisation and key management for authentication, respectively. And BCT-based authorisation helps to update specific fields of tokens to retain the communication ratio by reducing vehicle-to-vehicle losses [93,99,100,101].

However, though the application of AIML provides greater opportunities to counter cybersecurity risks, its usage has limitations as well [91]. At present, AIML applications as countermeasures for dynamic and sophisticated cybersecurity defence systems remain mainly at the proof-of-concept stage. As of now, it is not clear how much development in general computing is penetrated for detecting cyberthreats [92]. Also, the major issues with the usage of AIML in cybersecurity include the lack of AIML’s full autonomy, data privacy risks, insufficient legal frameworks, and ethical concerns arising from autonomous decision making and its adversarial usage due to its dual use. It can be used for defensive and offensive purposes as well [94]. Cybersecurity will be governed by regulations, while hackers have the freedom to use AIML technology for developing and introducing multiple quick, accurate, efficient, advanced, and vicious malicious attacks to break the IOMV system, including AIML-enabled security tools as well; e.g., a cyber-attacker can create an identical or fake website resembling a legitimate website and may be able to trick AIML-based security systems by spoofing it as a real one [98]. Also, AIML systems are not always perfect, as they can make mistakes by incorrectly flagging an activity as malicious due to incorrect data labelling or the overfitting of training data. Or AIML can be biased due to skewed training data, etc.; therefore, even AIML techniques may not provide protection from all types of threats. And AIML techniques need to be redesigned, updated, and maintained, as hackers keep on improving and updating the cyberthreats. Owing to the fast adoption of AIML in cybersecurity, it is necessary to resolve these related risks and concerns at a faster pace [94,98]. 

AIML is used to analyse data at runtime and to detect unauthorised users in the early stage, especially for tracking online harm. As of now, systems solely using AIML techniques are not completely secured, because the encryption techniques have some loopholes, such as the algorithm’s short life expectancy and low computation power, the use of various AIML techniques for various categories of faults, etc. [92,94]. Therefore, future cybersecurity may use artificial general intelligence: e.g., to analyse the behaviour of the intruder and overcome encryption weaknesses, artificial general intelligence (AGI) can be used to detect the rational behaviour of the different attack types. Human intelligence cannot apply encryption techniques but uses different cognitive correlations, like intention, perception, motivation, emotions, and implicit and explicit knowledge, to keep sensitive information confidential. Therefore, using AGI techniques can help to emulate human-like rationality for protecting information like a human mind would [103,113]. Also, like the application of AGI, the concept of Explainable Artificial Intelligence (EAI) seems to possess the potential to revolutionise cybersecurity in IOMV systems by better understanding the behaviour of various types of cyberthreats and allowing the development of effective countermeasures. The EAI method can become the most suitable for IOMV systems, where peer-to-peer communication takes place between healthcare devices, and anomaly-based intrusion detection in the IOMV networks [114,115]. Also, brain-inspired neuromorphic computing techniques are promising biologically inspired methods by using brain cognition mechanism to address diversified technologies for IOV system to develop intelligent and fault-tolerant transport systems [116].

The analyses of the research indicate that, to develop these new security enhancement strategies, a few techniques, like machine learning and artificial intelligence (AIML) with big data like blockchain technology (BCT) [60], can be promisingly employed to develop anomaly-based intrusion detection systems [81] and to defend against cybersecurity attacks. And a Cognitive Radio Network [69,117] offers an innovative way of solving spectrum underutilisation problems by exploiting spectrum opportunities like spectrum prediction, monitoring, and MAC protocols, to mention a few proposed solutions, but their holistic applicability must be determined as the research continues to progress in this field. 

### 5.3. Assessment of Cybersecurity Risk in IOMV System

As per the NIST Cybersecurity Framework, “risk of cybersecurity is an effect of uncertainty regarding information & technology, which may be caused due to loss of confidentiality, integrity, availability of information, data, control systems and which can potentially impact organisations’ operations”. And this NIST Framework defines how to assess cybersecurity risks and how to prioritise the actions to reduce those risks. Also, another standard framework, ISO-27001:2022, defines information security management, risk assessment, and risk treatment [39]. But all cybersecurity risks may not be equal. Therefore, the cybersecurity risk is subject to identifying security threats, the degree of vulnerability, the likelihood of exploitation, severity, frequency of occurrences, and their impact. And it can be estimated based on quantitative and qualitative assessments. The cybersecurity risk can be quantified in a mathematical equation, as in (2). Various types of potential cybersecurity attacks in IOMV systems are provided in Table 11.
Cybersecurity Risk = (Threat) × (Vulnerability) × (Information Value)(2)

Similarly, as shown in Table 12, the assessment of the qualitative cybersecurity risk helps to identify the likelihood of occurrence and its impact as per the Risk Assessment Matrix (RAM). The qualitative risk assessment is estimated based on the combination of the likelihood and severity. For example, with a combination of *x*_11_ and *x*_21_, the risk is low. Similarly, with a combination of *x*_24_ and *x*_34_, the risk is considered critical. Based on this degree-of-risk estimation, the risks can be prioritised, and accordingly, countermeasures specific to the type of risks need to be taken. 

The approach to the management of cybersecurity risk needs to be dynamic. The steps involved in the mitigation of cybersecurity are shown in the flowchart in Figure 19, which specifies the levels of acceptance in assessing a risk. Then, a suitable risk assessment strategy is chosen. The risks are prioritised based on their degree of impact and importance. Then, control strategies and countermeasures are developed and implemented to strengthen cybersecurity in the IOMV.

### 5.4. Parameterisation of Cybersecurity

A “parameter” is a value or a metric that is used by a function to control the operation or to determine or compute the outputs of a function. And the mathematical representation, in terms of a parameter for expressing the state of a system, a process, or a model as a function of any independent quantity, can be termed the “process of parameterisation”.

In the assessment of cybersecurity risks, the key parameters that can be used to determine the vulnerability, the strength of cybersecurity, system resilience, etc., include the MTTI (Mean Time to Identify) and MTTR (Mean Time to Respond). These parameters provide the degree of vulnerability of the cybersecurity of the IOMV system and how much time it takes to identify a potential cyber-attack and then respond to such threats in the network or system. The higher the values of MTTI and MTTR, the greater the loss to the system. The MTTR is a direct measurement of how long critical systems of the IOMV remain offline due to an attack. As shown in Figure 20, there are other metrics that are within the breach-response timelines, which can help to estimate the degree of cybersecurity of the IOMV system. They are the MTTR (Mean Time to Repair), MTBF (Mean Time Between Failures), MTTF (Mean Time to Failure), MTRS (Mean Time to Restoration), and MTBSI (Mean Time Between System Incidents). These common parameters help to measure each segment of the cybersecurity breach timelines from the initial penetration to recovery to the next attack. These numbers help to understand the effectiveness of the IOMV cybersecurity system.

Referring to Figure 20, the MTTD is the average time duration that the system takes to notice a breach from the time the actual breach has taken place. And the MTTI is the average time duration that the system takes to identify the breach from the time it has actually noticed the breach. The MTTR is the average time that the system takes to fix the breach of the system from the time the system has actually noticed the breach. MTTRepair is the average time that the system takes to bring back the system to normalcy, i.e., the average time that it takes the system to repair the breach and restore it to normalcy from the time the actual breach in the system is noticed. The MTBSI is the average time between breaching incidents from the time the previous breach was noticed. And the MTBF is the average time between when the system was made available after the previous breach and the new breach. For the strong cybersecurity of an IOMV system, parameters like MTTD, MTTI, MTTR, MTTRepair should be very small, and parameters like the MTBSI and MTBF are ideally maximized. These can be achieved by strong countermeasures in combating cyber-attacks.

## 6. Results 

The aim of the present work is to provide guiding information to researchers seeking to investigate and analyse subjects related to “cybersecurity in the IOMV”, an IOT-based application that is derived from the integration of IOT applications, i.e., IOV and IOMT, and can help to enhance connected healthcare services as part of developing smart-city realm. This is realised by making innovations and merging physical infrastructure. This article, based on our systemic state-of-the-art analysis, provides suggestions for future research on cybersecurity attacks, gaps, factors influencing cybersecurity, and challenges to overcome in implementing defence strategies, etc., on the subject/topic of “Cybersecurity in Internet of Medical Vehicles”. The following are the major findings and outcomes of our analysis based on the research work presented in this paper related to the “Cybersecurity in Internet-of-Medical-Vehicles”:We perceive that this is a niche yet unexplored research area and that it appears to be an upcoming and widely trending topic, as the concept of the IOMV is basically derived from integrating IOT-based applications, i.e., the IOV and IOMT.Most of the research gaps corresponding to this topic of the IOMV are still unexplored, and there is a great need to implement robust cybersecurity measures due to increasingly sophisticated cyber-attacks; many challenges and factors influencing security vulnerabilities, like integrity, interoperability, etc., and many issues like few defined standards, regulations, guidelines, frameworks, protocols, etc., are significant gaps and issues that need to be explored and addressed, and hence, it is becoming an upcoming and widely trending topic.Also, as part of the study and analysis, we have presented various types of potential cyber-attacks that can take place at different layers of the IOMV system.We have discussed and provided the application of AIML and AGI techniques as solutions for developing countermeasures to strengthen cybersecurity in IOMV systems.We have presented the IOMV system model and architecture and the types of communications that can take place at different layers of the IOT structure.We have presented details of relevant standards, protocols, frameworks, and guidelines applicable to the IOMV.Factors affecting the cybersecurity risk in the IOMV system have been discussed, along with various challenges and constraints affecting the cybersecurity risk in the IOMV.We have presented methods for assessing the cybersecurity risk and parameterisation details for assessing the strength of cybersecurity, which can help to develop countermeasures for cybersecurity.And we have discussed the future scope to focus on, which can be helpful for researchers to develop robust cybersecurity in IOMV systems.

## 7. Limitations 

This paper is primarily about discussing a state-of-the-art analysis regarding the research topic of cybersecurity in the IOMV, which is a derived concept based on the integration of IOT, IOV, and IOMT applications. Therefore, only major journals on diverse topics from the vast literature are considered for discussion to gain knowledge, to know the various visualisation tools used and types of analyses performed in those papers, and to understand the objectives, research challenges, and limitations of their studies. As the safety and security of the IOMV system is a broad area, admittedly, this work may not have fully or thoroughly covered all the security aspects of the IOMV system. This paper discusses and analyses various kinds of cyber-attacks and techniques for strengthening cybersecurity in the IOMV. As part of our future research works, to understand the behaviour of the IOMV system, these cyber-attacks need to be evaluated by performing simulations in terms of performing risk assessments and parameterisation. And then, various solutions for countering the cybersecurity risk need to be developed. Also, as the focus of this paper is to discuss the state-of-the-art analysis only, proposals of new methods or algorithms for countering cyber-attacks and strengthening the cybersecurity of the IOMV shall be part of our future research works. 

## 8. Challenges and Constraints

The applications of IOT technology are vast and virtually have no limits, and they are rapidly growing—both in size and in scope in different fields. The rapid evolution of the IOV and IOMT and thus the IOMV system, which is the integration of applications of the IOT, i.e., IOV and IOMT, helps to shape the infrastructure of “connected-healthcare” services as part of the “smart-cities” realm. However, the corresponding industries, including the automobile and healthcare industries, need to prepare for different types of constraints and challenges during their integration and the associated cybersecurity risk of the IOMV system. The following are some of the major constraints and challenges.

### 8.1. Unexplored and Niche Area 

As the applications of IOT technology in the fields of IOV and IOMT are themselves new, the combined application derived from integrating IOV and IOMT, i.e., the IOMV, the “Internet of Medical Vehicles”, is a further new, futuristic, and yet still unexplored research area. 

### 8.2. Not Immune to Variety of Security Anomalies

The issue of cybersecurity risk is one of the primary yet critical and difficult parts of all the challenges in the deployment of the IOMV system that need to be addressed by the automobile and medical industries, because it is more vulnerable to cyber-attacks, as it involves dealing with sensitive health data and health-monitoring systems [118,119,120] in a vehicle application, and data breaches and unauthorised access can occur due to exposure to various communication systems and interconnected networks, very limited computing resources, and the typical lack of traditional security controls; these systems are not designed to be immune to the variety of security anomalies.

### 8.3. Complicated Requirements of IOMV System

The implementation of cybersecurity measures in the IOMV is complicated and yet still not explored, as it is subject to various factors and constraints because potential cybersecurity risks in the IOMV can take place with different types of attacks at various layers right from the source, i.e., onboard connected medical devices, up to the target recipient devices at the hospitals and their interconnecting communication channels and networks. Those significant factors and constraints are due to the requirements for:Seamless Integration—with existing technology and infrastructure.Scalability—without any significant decline in IOMV system performance, scalable and flexible architectures are needed to maintain uniformity in handling the future demand of rapidly increasingly connected devices, as scalability affects network capacity and device and data management, e.g., healthcare devices and their increased data and communication volume.Interoperability—connecting many diverse devices.Usability and Cost—tradeoff and balance between the cost of deployment and benefits.Reliability—performance of the system can suffer even if one IOT device is unresponsive or compromised due to network blockages, DOS attacks, and malfunctions in communications.Power Management—device-level energy issues, as IOT devices are developed to be smaller, use low power, and operate using batteries.Data Management—collection, storage, and processing of data.Lack of Standardisation—absence of common agreed-upon specifications, which leads to inefficient interconnection, communication, and exchange of data between devices.Lack of Protocols—in the automotive and healthcare industries, systems manufactured by various manufacturers follow their own standard rules and protocols.Lack of Skill Set—conventional vehicle manufacturers/operators and medical personnel are not trained for cybersecurity adversaries, and usually, IT systems are supported by the IT teams.Device and Network Infrastructure—continuously building, maintaining, and supporting connections of large numbers of advanced IOT devices used in IOV and IOMT applications require regularly updating existing devices.Poor Connectivity—IOT sensors are required to monitor process data and supply information.

### 8.4. Complex Nature of Cybersecurity of IOMV System

There are many significant issues that make providing cybersecurity in the IOMV system difficult. The main issues include:Lack of encryption—where hackers can easily manipulate the software.Inadequate testing and updating—automotive and medical manufacturers not placing much emphasis on system testing and updating, showing eagerness to manufacture and deliver their devices into markets.Brute-force password and weak credential details—make the IOT devices used in the IOV and IOMT vulnerable to cyber-attacks.Software Vulnerabilities—can happen from errors and mistakes like bugs and weaknesses in the software code by using unsupported or outdated software that can be exploited by attackers to carry out malicious attacks.

The various issues, as discussed in Section 8, explain the significant reasons that cybersecurity risks arise. A cybersecurity risk can also take place at various points and layers of the IOMV system, from the source of data-generating IOT devices, through the networks and communication channels, to the target recipients [121,122]. And the typical significant types of cyber-attacks, in which the attackers can exploit or hack the IOMV systems, include attacks that can happen against IOT devices; a few are specific to IOMT devices and a few are related to the IOV application. And the majority of those attacks are common, and a few are specific to either the IOV level or the IOMT level. And a few of them are more vigorous due to their undetectable nature, and a few others have the potential to create disruption or completely shut down the IOMV system.

Therefore, the above challenges and constraints only highlight the complexity of the cybersecurity risk and finding countermeasures in the IOMV application. All of the challenges and constraints, as discussed above, along with the different types of cyber-attacks, are interrelated, affecting cybersecurity and the overall performance of the IOMV system.

## 9. Future Outlook

Researchers estimate that the number of “connected things” part of the IOT will reach more than 40 to 50 billion by the year 2030. The applications of “IOT in Vehicles” and “IOT in Medical Things” are two distinct fields. Together, they represent a higher proportion of the IOT network along with their interconnected sensors, applied devices, communication systems, various integrated technologies, standards, services, etc., and their applications as the IOV and IOMT are only projected to grow rapidly as they help to shape the infrastructure of the smart-city [87] realm, offering safer, autonomous, connected, and intelligent transportation and meeting the increased need for cost reductions in connected healthcare services, respectively, which is influencing the growth of the IOV and IOMT markets. 

The automotive and medical industries are at an important juncture in history. Though the concept of the connected car is offering an exciting new era of car ownership to drivers, this expanded capability introduces cybersecurity risks that could threaten the safety of drivers. Cybercriminals are becoming more sophisticated in their attacks, and the types of cyber-attacks in the IOT system are manifold, as illustrated in Table 11, and day by day, their numbers are alarmingly increasing, e.g., IOT malware attacks rose by 220% in 2022 as compared to 2020. The automotive industry witnessed a sharp increment of approximately 600% in incidents related to cybersecurity from 2016 through 2019, and for the IOMT, the increase was about 405%. It is becoming a serious concern that, because of the growth in the rate of attacks, combined with the growing number of IOT devices, the combinations of hacking or methodologies of hacking are becoming sophisticated. Therefore, the risks of cyber-attacks are evident and real, which needs to be addressed, and cybersecurity in the “Internet of Vehicles” [123] and “connected medical devices” [23,124] is a fast-expanding research field that interests serious researchers and industries. 

The “cybersecurity in IOMV”, i.e., the cybersecurity risk arising after the integration of the IOV and IOMT, is an important, futuristic, and need-of-the-hour topic, which is crucial to the concept of providing connected healthcare services as part of the development of smart cities [87,125,126]. But, at present, there is no single technology or one-stop solution that can be deployed to prevent the multitude of types of cyber-attacks and cybersecurity risks in the IOMV. And due to a lack of regulations, standards, and protocols, and as various devices are interconnected with multiple communication technologies and communication points, the security and privacy of these interconnected networks still have potential risks [47].

Human security experts may not be capable of handling the degree of security threats. And they must have strong support from intelligent machines, software, and the latest technologies for fighting cyberthreats. Cybersecurity capabilities can be enhanced by the usage of AIML techniques. They can comprehend and simulate human intelligence. AIML techniques can improve the overall security performance, as they are more flexible, adaptable, and robust. But their usage has limitations as well. At present, AIML applications as countermeasures remain mainly at the proof-of-concept stage, as they lack full autonomy and are associated with data privacy risks, insufficient legal frameworks, and ethical concerns. Also, AIML techniques can be misused by hackers to develop and introduce multiple quick, accurate, efficient, advanced, and vicious malicious attacks to break the IOMV system, including AIML-enabled security tools as well. And the systems solely using AIML techniques are not completely secured due to the algorithm’s short life expectancy, low computation power, etc.; therefore, future cybersecurity may use artificial general intelligence and Explainable Artificial Intelligence, which have the potential to revolutionise cybersecurity in IOMV systems. They can be used to detect the rational behaviour of different attack types. They can help to emulate human-like rationality and cognitive correlations like intention, perception, motivation, emotions, and implicit and explicit knowledge to keep sensitive information confidential.

Several challenges and implications, as discussed, exist today that need to be addressed before the mass adoption of the IOMV, and such cybersecurity enhancement measures need to satisfy balancing the trade-off of their deployment and the cost of implementation. This can be achieved through collaborations among researchers, relevant industries, various organisations, and standardisation bodies. The security and privacy of these interconnected networks still have potential risks [47] and addressing these issues [33] of cybersecurity is the need of the hour and has enormous potential for research at various levels [127] of interconnected networks [29].

This paper discusses and analyses various kinds of cyber-attacks and techniques to strengthen cybersecurity. In the future, they need to be evaluated by considering and simulating cyber-attacks, performing estimations, updating parameterisation, etc., to ascertain the behaviour of the IOMV system in terms of countering the cybersecurity risk.

## 10. Conclusions

This analysis focuses on an upcoming and futuristic trend, “Cybersecurity in Internet of Medical Vehicles”, which is a niche yet unexplored research area. This analysis was performed based on the published literature between 2016 and 2023 from the databases “Scopus” and “Web-of-Science”. The main objectives of this journal paper regarding this research field are to evaluate global trends in terms of top publication outputs, publication patterns, gaps, future outlooks, etc. The analysis presented in this paper will help beginners, relevant industries, and serious researchers to distinguish the topmost authors and their contributions to the research work. Based on our analysis, we found that “cybersecurity” is an extensively used keyword. The privacy and security of medical data related to onboard patients are critical in delivering “connected-healthcare” services using the IOMV system as part of the smart-city realm. Connected IOT devices of the IOMV have to offer exact data and should be unaffected by noise factors. The “Internet-of-Medical-Vehicles” has security risks from a variety of sophisticated cyber-attacks as devices communicate with different networks, and this can endanger the onboard patient’s life. Hence, it is critical to understand subjects related to “cybersecurity” in these vehicles to develop robust cybersecurity measures. And our analysis in this paper will help researchers to gain a comprehensive idea of this multidisciplinary research area related to “Cybersecurity in Internet-of-Medical-Vehicles”, as it presents a review of top journals, highlighting their gist, merits, and limitations, an analysis of highly cited papers, the IOMV architecture and system model, related applicable standards, types of potential-cyber-attacks, factors causing cybersecurity risks, various artificial intelligence techniques as solutions for developing potential countermeasures, the assessment and parameterisation of cybersecurity risks, constraints and challenges, and future outlooks for implementing cybersecurity measures in the IOMV. Also, our analysis reveals that there has been a sudden growth in publications in this research area since 2018 and that this is a niche yet unexplored research area with few defined standards, regulations, and frameworks, and there is a great need to implement robust cybersecurity measures due to increasingly sophisticated cyber-attacks. And we found that very little work has been conducted in this area of research; therefore, there are great opportunities for researchers to explore and work towards contributing to this upcoming, futuristic research field. 

## Figures and Tables

**Figure 1 sensors-23-08107-f001:**
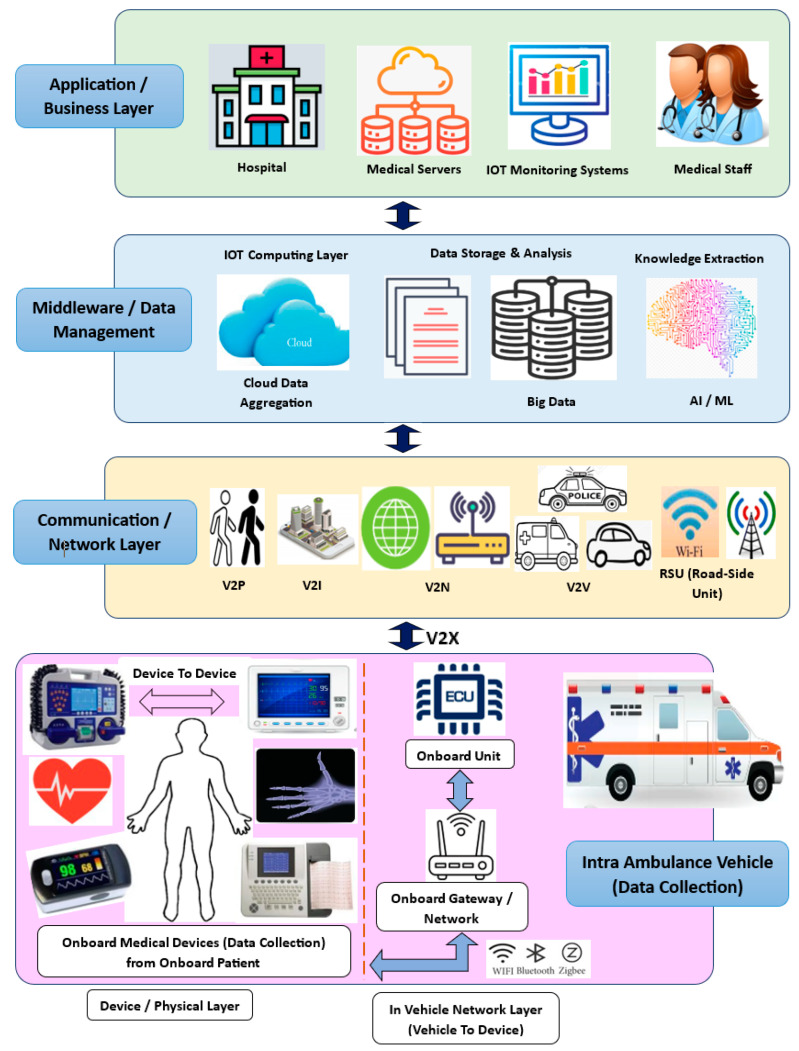
Architecture of Internet of Medical Vehicle (IOMV) system.

**Figure 2 sensors-23-08107-f002:**
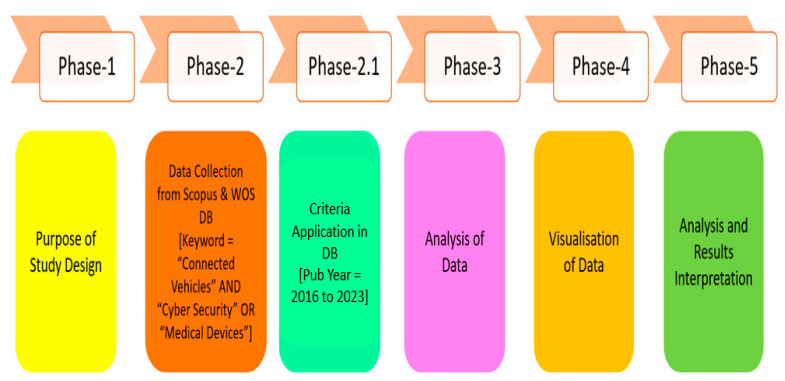
Step-by-step research methodology process.

**Figure 3 sensors-23-08107-f003:**
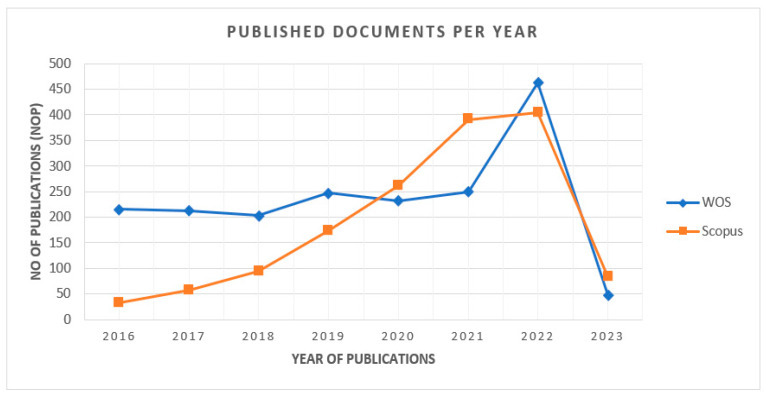
Total NOP between 2016 and 2023 in DBs Scopus and WOS.

**Figure 4 sensors-23-08107-f004:**
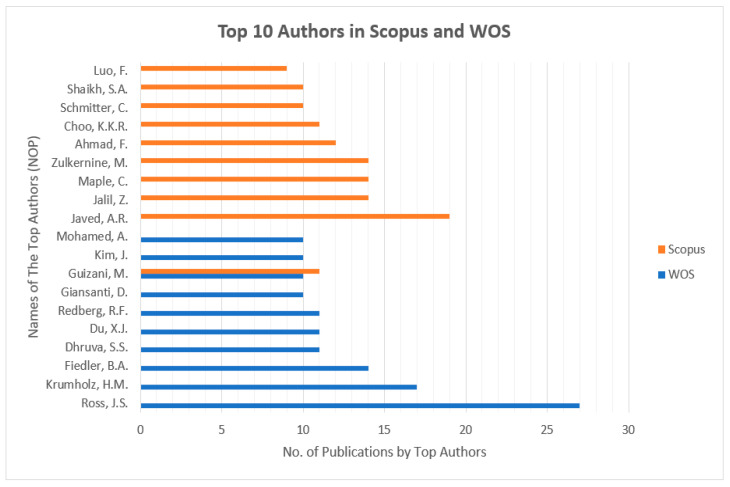
Top-10 authors from DBs Scopus and WOS, along with their NOPs.

**Figure 5 sensors-23-08107-f005:**
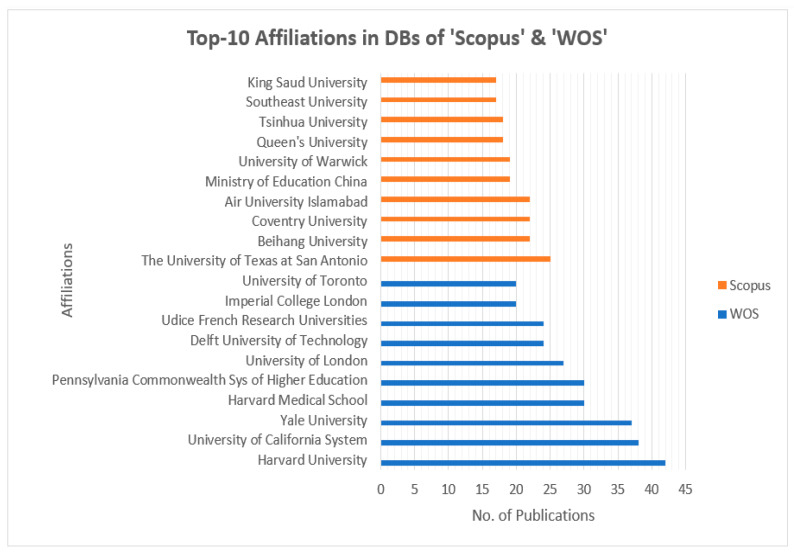
Top-10 affiliations from DBs Scopus and WOS, along with their NOPs.

**Figure 6 sensors-23-08107-f006:**
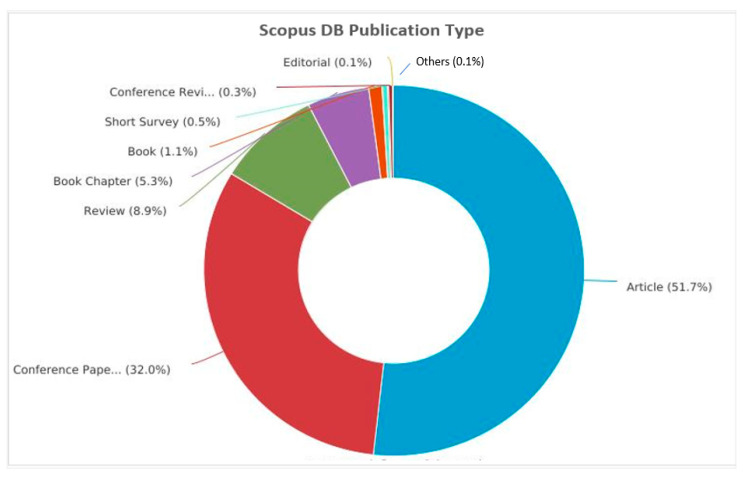
Top-10 types of documents published from 2016 to 2023 in Scopus DB.

**Figure 7 sensors-23-08107-f007:**
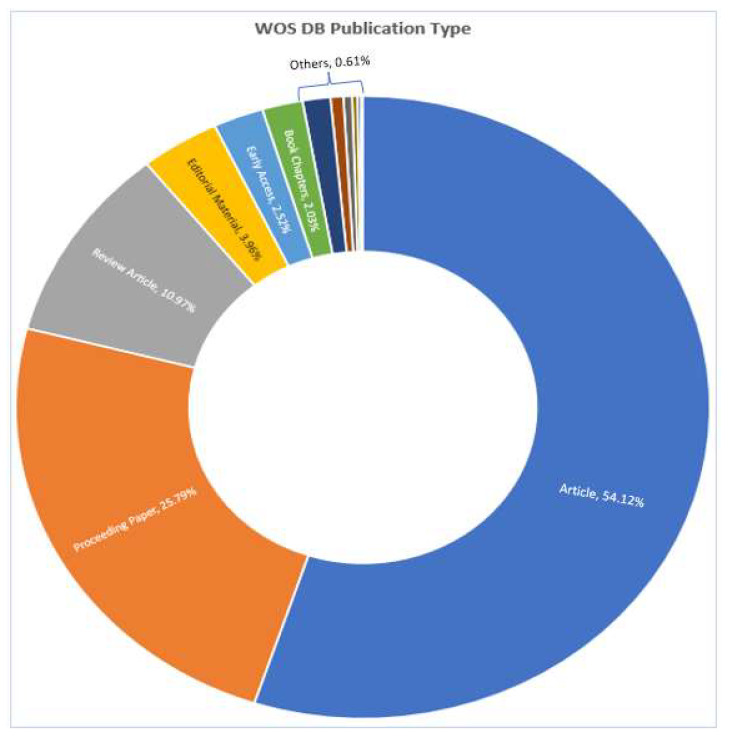
Top-10 types of documents published from 2016 to 2023 in WOS DB.

**Figure 8 sensors-23-08107-f008:**
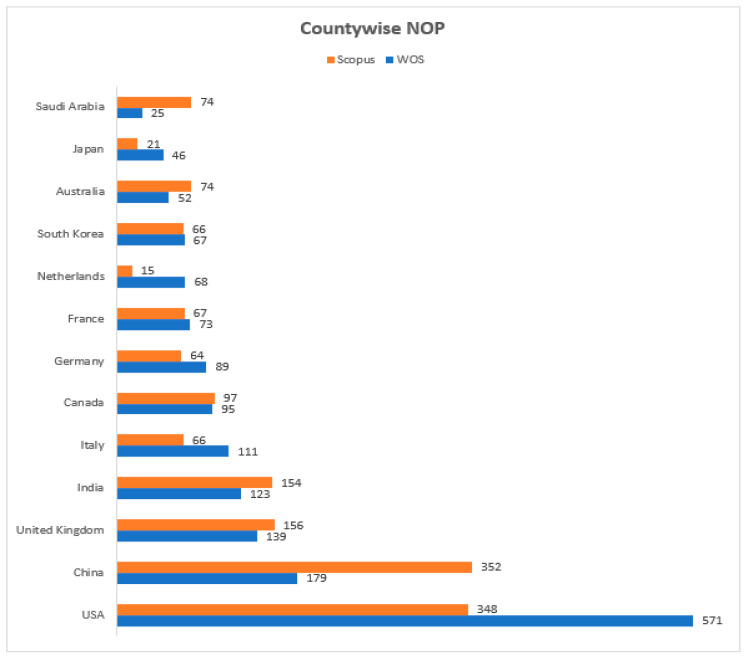
Country-wise No. of Publications (NOP) in Scopus and WOS DBs from 2016 to 2023.

**Figure 9 sensors-23-08107-f009:**
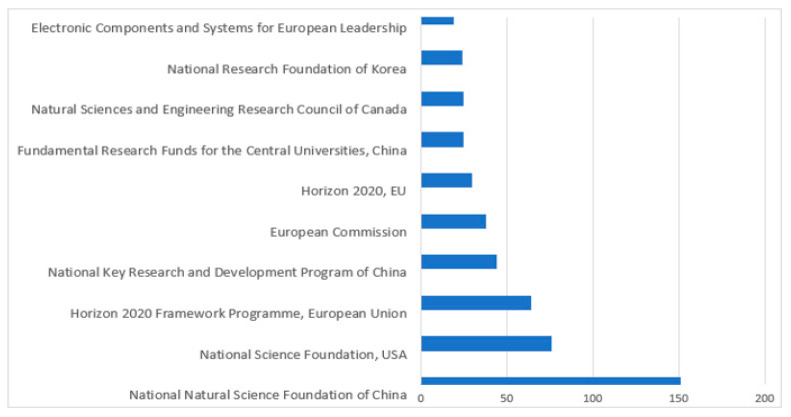
Top-10 sponsoring agencies from Scopus DB publications from 2016 to 2023.

**Figure 10 sensors-23-08107-f010:**
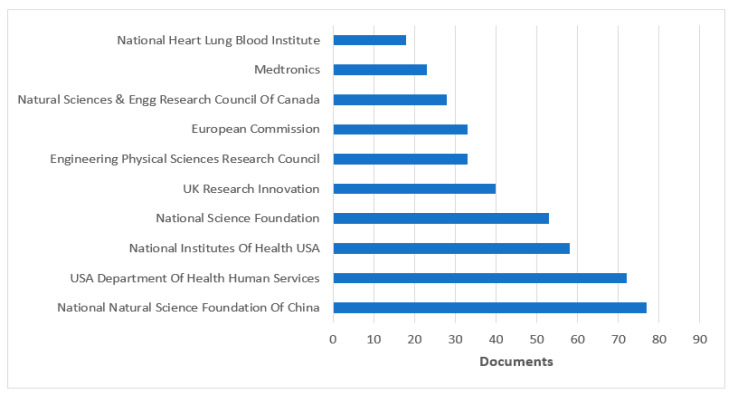
Top-10 sponsoring agencies from WOS DB publications from 2016 to 2023.

**Figure 11 sensors-23-08107-f011:**
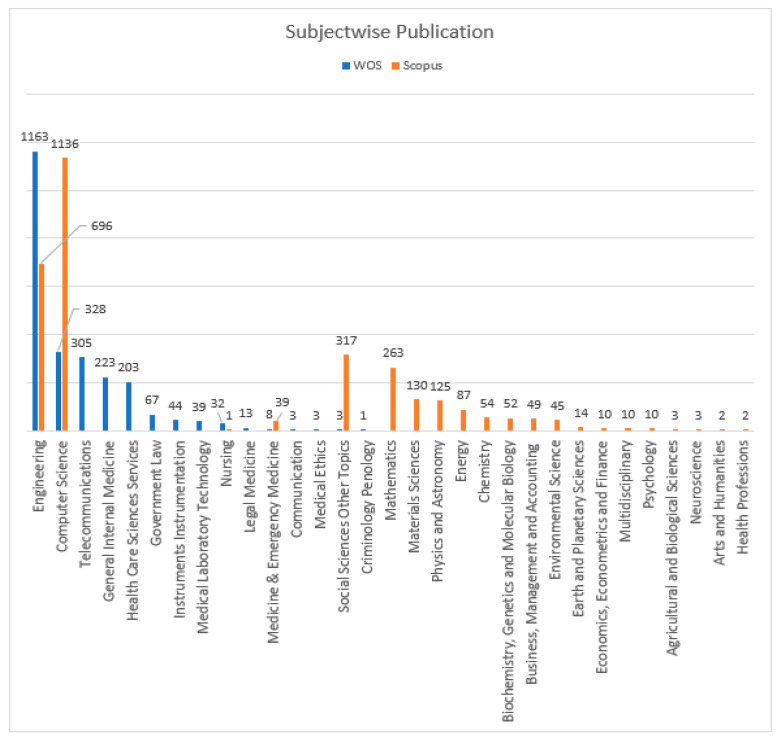
Subject-wise “NOPs” in “Scopus” and “WOS” DB from 2016 to 2023.

**Figure 12 sensors-23-08107-f012:**
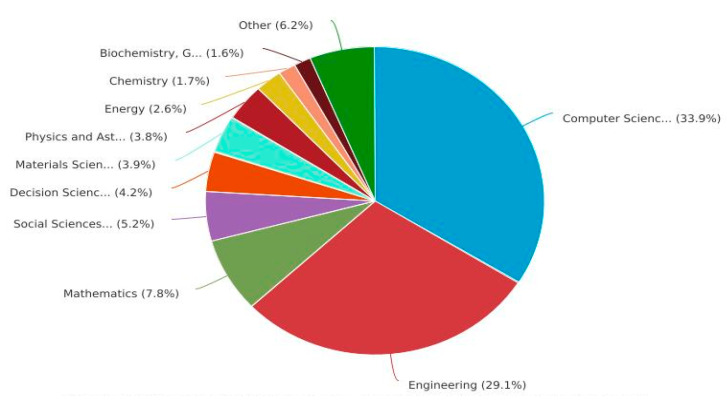
Percentage of documents published with subject areas in the DB Scopus.

**Figure 13 sensors-23-08107-f013:**
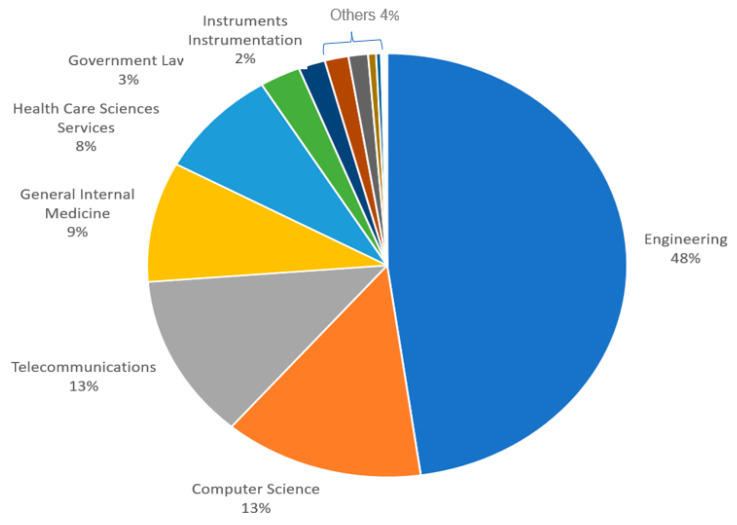
Percentage of documents published with subject areas in WOS DB.

**Figure 14 sensors-23-08107-f014:**
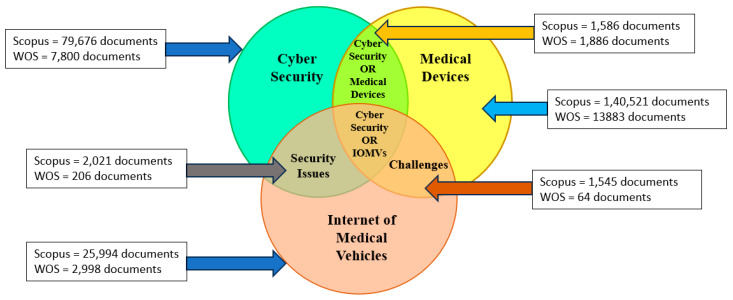
Critical analysis of top-cited paper of Scopus and WOS DBs.

**Figure 15 sensors-23-08107-f015:**
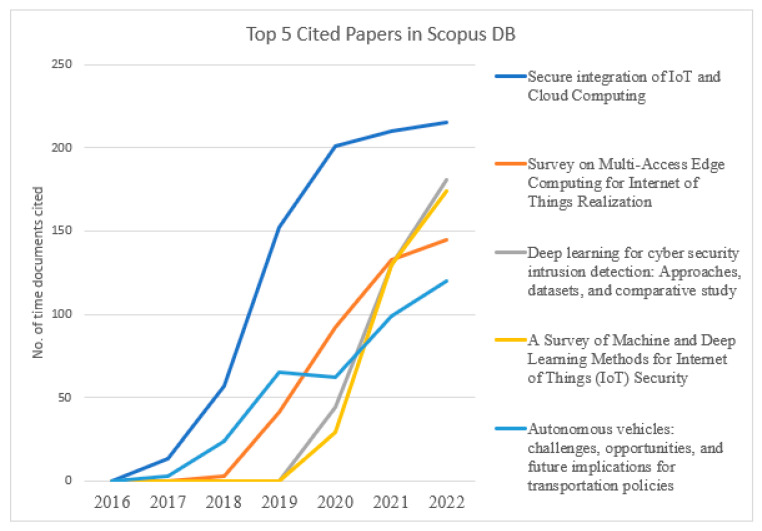
Top-five cited papers on cybersecurity in IOMV over five years in Scopus DB.

**Figure 16 sensors-23-08107-f016:**
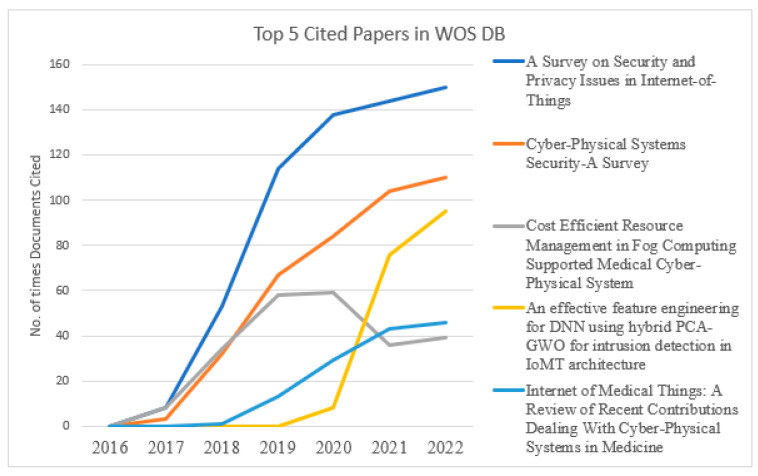
Top-five cited papers on cybersecurity in IOMV across five years in WOS DB.

**Figure 17 sensors-23-08107-f017:**
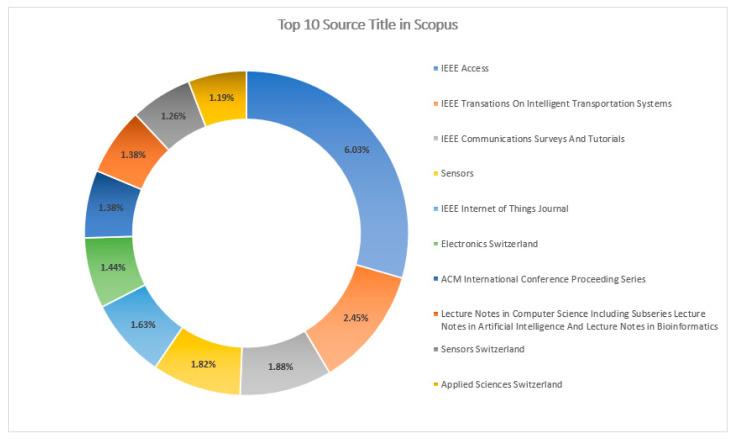
Top-10 source titles in Scopus DB.

**Figure 18 sensors-23-08107-f018:**
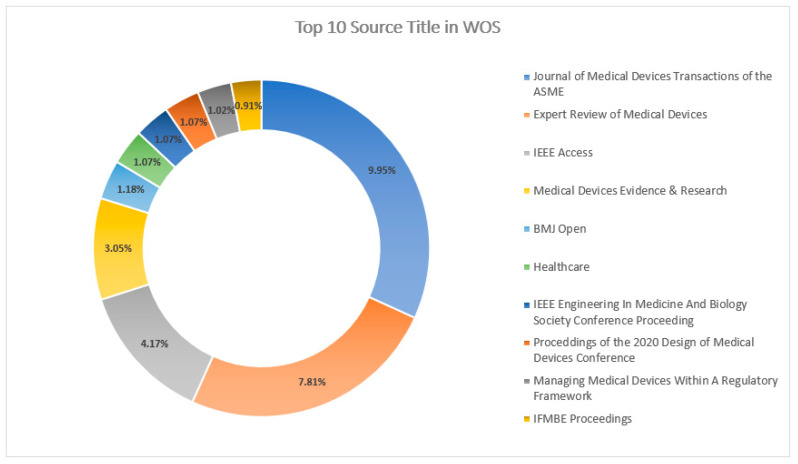
Top-10 source titles in WOS DB.

**Figure 19 sensors-23-08107-f019:**
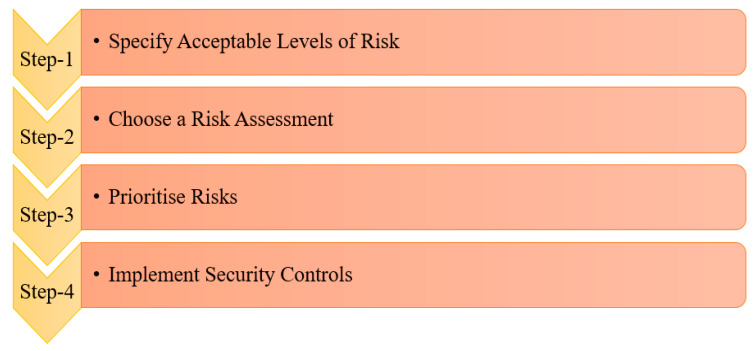
Flowchart showing the steps for addressing cybersecurity risk in IOMV.

**Figure 20 sensors-23-08107-f020:**
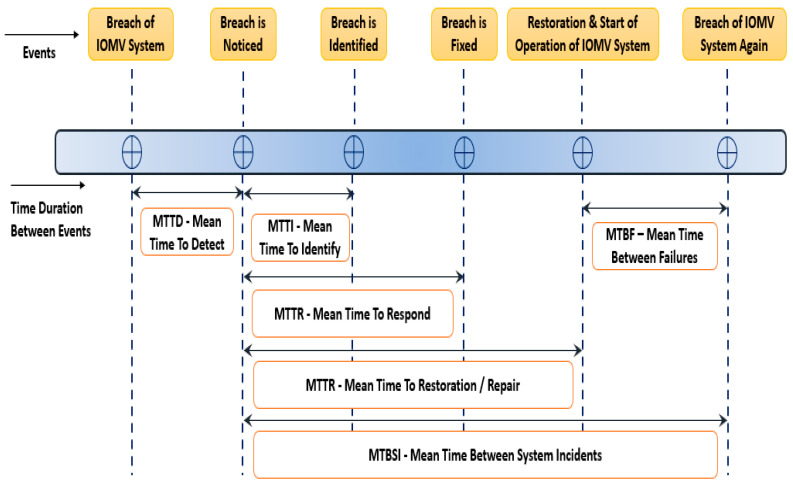
Parameters and metrics to measure strength of cybersecurity of IOMV.

**Table 1 sensors-23-08107-t001:** The top-15 keywords with the number of documents published from Scopus DB.

Keywords	Number of Publications
Network Security	424
Vehicles	340
Cyber Security	317
Cybersecurity	265
Internet of Things	255
Security	200
Autonomous Vehicles	180
Vehicle-To-Vehicle Communications	171
Computer Crime	141
Intelligent Systems	138
Intrusion Detection	124
Machine Learning	116
Embedded Systems	114
Blockchain	108
Intelligent Vehicle Highway Systems	105

**Table 2 sensors-23-08107-t002:** List of analysis studies regarding literature analytical methods and “Cybersecurity in IOMVs”.

Citations	Area of Study	Journal	Tools
[53]	Vision, applications, and future challenges of IOT	*Emerald Insight*	Force Atlas Layout, Gephi and Hitscite, BibExcel
[54]	Interaction between AVs and human-driven vehicles, safe and efficient operation of AVs	*Sustainability*, MDPI	VOSviewer, Biblioshiny
[55]	Autonomous vehicles, challenges in sustainable urban mobility	*Sustainability*, MDPI	VOSviewer
[56]	Healthcare and cybersecurity—gaps and opportunities	*Journal of Medical Internet Research*	Abstrackr, Leximancer, Heat Maps
[57]	Advancements in cybersecurity and information systems in healthcare	*International Conference on Intelligent Systems, Advanced Computing and Communication*	VOSviewer
[58]	Blockchain-based Internet of Medical Things	*Applied Sciences*, MDPI	WordCloud
[60]	IOT in healthcare, smart healthcare	*Internet of Things*	Gephi, BibExcel
[61]	Performance improvement with cloud connectivity of IOT-sensed devices and real-time data	*South African Institute of Electrical Engineers*	WordCloud
[62]	Cybersecurity trend analysis	*European Journal of Molecular & Clinical Medicine*	VOSviewer

**Table 3 sensors-23-08107-t003:** List of analyses corresponding to review journals relevant to “Cybersecurity in IOMVs”.

Review Doc Ref.	Discussed Points	Advantages	Disadvantages
[24]	Protection of medical sensors from harmful interference like EMI Robust power control using wireless network for e-health applications Data transmission framework over IOV for e-health purposes	Simulation and modelling for enhancing network capacity and performance Proposed power control algorithm reduces the amount of EMI on medical sensors	Analysis of dynamic setting is needed; i.e., the structure of the Internet of Vehicles changes dynamically over time.
[47]	IOT architecture and communication models Survey on 3-layer IOT architecture (Perception/Network/Application)	Communication technologies used in IOT, security attacks and their countermeasures Possible types of attacks at different layers of IOT system	Existing security worries in boundless heterogeneous networks need to be tackled. Design a lightweight, robust security system with data confidentiality, availability, and integrity.
[81]	Review of network-based intrusion-detection-system-based articles is provided with strengths and limitations of proposed solutions Comparative study of ML/DL techniques—network-based intrusion studies and their methods with merits/demerits of suggested methodologies	Recent trends and advancements of ML/DL-based NIDS are given (proposed methodology, evaluation metrics and dataset selection) Generalised ML/DL-network-based intrusion detection system method	Proposed schemes are complex. Need extensive computing resources, like processing power, storage capabilities. Research gaps in improving model performance for low-frequency attacks, absence of systematic dataset, lower performance in a real-world setting. Solutions with regard to complex models needed.
[67]	V2X communication and integration of sensor data of vehicles for autonomous vehicles Modular and scalable C-ITS architecture (in-vehicle sensor CAN bus data, fascia smartphone, car sensors with cloud connectivity, localised vehicular communications, and long-range mobile nets)	Implementation details in devices (Onboard Unit, OBD-II reader and smartphone) Tests conducted; results used to justify the intended system Data collection system can be a valuable tool for improving road safety	Proposed outcome is constrained due to non-standardised CAN messages (specifically need to configure for each vehicle/OEM) Obtained vehicular sensor data have many uses
[65]	Security issues in smart automobiles, their attacks and solutions.	Types of cyber-attacks with categorisation of security attacks in VANETS and IOT and different threats	Efficiency of information routing and dissemination (e.g., multiple MAC-address-based issues in multi-level comm. that take place during transportation need efficient comm. protocols in VANETS)
[7]	Various aspects of connected vehicles (CVs); their effects on transport system and urban mobility. Short-range wireless communications, Veh-to-Veh (V2V) and Veh-to-Infra (V2I). Evolution of CVs, operational aspects with its applications	Impacts, operational benefits of CVs Review of CVs’ best practices and initiatives Investigations of public perception of CV applications Challenges to CV technology are identified.	Further study is required on AVs to verify tradeoff amongst capacity improvement and occupant comfort and to fully know mobility data needs Performance effort, replacing classical vehicles with CVs and needed infrastructure
[20]	Identification, comparison, and classification of existing assessments in healthcare IOT systems 5 categories classified (sensor/resource/communication/application and security-based approaches)	Benefits/constraints of selected methods; contrast in terms of evaluation—techniques, tools, metrics	Consider open issues such as power management, trust, and privacy, fog computing, multi-objective optimisation, resource management, blockchain, tactile Internet, QOS, SDN/NFV, big data analytics, online social networks Interoperability, mobility, scalability, and real-testbed implementation needed
[28]	Exploration of the latest research in IOT security (2016 to 2018); its developments and open problems.	Overview of the existing condition of IOT-security research, IOT modellers and simulators, relevant tools	Necessity for development of complete IOT threat modelling, design of a zero-trust algorithm to mitigate known and unknown cyber-attacks in IOT systems
[27]	Cybersecurity state in IOT domain along with its security challenges	Security requirements and techniques to overcome challenge. Blockchain technology advised for IOT security	Required evaluation of blockchain integration with IOT, dealing with real-life models regarding security and privacy of IOT
[16]	IOT in health application as traditional medical practices unable to address demands of specific healthcare needs in today’s world setting Combine IOT and telemedicine for smart healthcare	IOT concept, its architecture, different layers and their processing, and communications systems in remote healthcare services	Still, the concept of IOT is not yet mature Telemedicine needs a fast, safer, and firm mobile comm. network for achieving objectives of real-time communications (between doctors and patients)

**Table 4 sensors-23-08107-t004:** Keywords used for querying DBs Scopus and WOS.

Keyword Type	Database	Keyword Combination
Primary keywords	Scopus and WOS	“Internet of Vehicles” AND “Cyber Security” OR “Medical Devices” OR “Internet of Medical Things”
Secondary keywords	Scopus and WOS	“IOT in Vehicles” OR “IOT in Medical Things” OR “Internet of Things” AND “Security in IOT” AND “Cybersecurity in Medical Things”

**Table 5 sensors-23-08107-t005:** Tips to search relevant document results in different ways using the keyword search.

Technique	Sign	Specimen Instance
Boolean	AND	All search terms must occur to be retrieved, e.g., Soft Drink AND Cold Drink
Boolean	OR	Any one of the search terms must occur to be retrieved, e.g., Soft Drink OR Cold Drink
Boolean	NOT	Excludes records that contain a given search term, e.g., Vehicle NOT Heavy. It retrieves documents with the vehicle term that do not include heavy.
Proximity		Search for an exact phrase and input the term in quotation marks. The use of quotation marks disables the lemmatisation of terms, e.g., “Cyber Security”. It shows documents related to the Cyber Security term only.
Truncation	$	Search for zero or more characters in between letters, e.g., hono$r = honor, honour
Truncation	**	Search for zero or more characters in suffix/prefix, e.g., *form* = formation, form, transform, inform
Truncation	?	Search for one character only in between letters, e.g., gre?t = great, greet

**Table 6 sensors-23-08107-t006:** Descriptive statistics of top-10 nation-states overall and countries with highly cited papers.

Regions	Max Publication Country-Wise	Countries with Highly Cited Papers	Country-Wise Highly Cited Publications
USA	571	China	56
China	179	India	8
United Kingdom	139	Australia	8
India	123	Canada	6
Italy	111	Turkey	4
Canada	95	Singapore	2
Germany	89	USA	2
France	73	Iran	2
Netherlands	68	Turkey	2
South Korea	67	Norway	2

**Table 7 sensors-23-08107-t007:** Descriptive statistics of top-10 nation-states overall and nations with highly cited papers.

One-Sample *t*-Test			
	*t*	*df*	*p*
Max Publications Country-wise	2.02	9	0.07
Highly Cited Publications Country-wise	1.75	9	0.12

**Table 8 sensors-23-08107-t008:** Descriptive statistics of top-10 nations overall and nations with highly cited papers.

Descriptive Statistics				
Stats	Max Publications Country-Wise	Highly Cited Publications Country-Wise	Nation-States	Highly Cited Papers Nation-States
Valid	10	10	10	10
Mode	5.00	1.00		
Median	5.00	1.50		
Mean	13.50	4.60		
Std. Deviation	21.06	8.31		
Skewness	2.89	3.03		
Std. Error of Skewness	0.68	0.68		
Minimum	3.00	1.00		
Maximum	72.00	28.00		
25th Percentile	4.25	1.00		
50th Percentile	5.00	1.50		
75th Percentile	12.75	3.75		

**Table 9 sensors-23-08107-t009:** Top-15 highly cited papers in Scopus DB for “Cybersecurity in Internet of Medical Vehicles”.

S.No	Publication Year	Publication Title	Authors	Journal Title	Cited by
1.	2018	Secure Integration of IoT and Cloud Computing	Stergiou, C., et al.	*Future Generation Computer Systems*	739
2.	2018	Survey on Multi-Access Edge Computing for Internet of Things Realization	Porambage, P., et al.	*IEEE Communications Surveys and Tutorials*	411
3.	2020	Deep learning for cyber security intrusion detection: Approaches, datasets, and comparative study	Ferrag, M.A., et al.	*Journal of Information Security and Applications*	405
4.	2020	A Survey of Machine and Deep Learning Methods for Internet of Things (IoT) Security	Al-Garadi, M.A., et al.	*IEEE Communications Surveys and Tutorials*	383
5.	2016	Autonomous vehicles: challenges, opportunities, and future implications for transportation policies.	Bagloee, S.A., et al.	*Journal of Modern Transportation*	356
6.	2019	A survey of machine learning techniques applied to software-defined networking (SDN): Research issues and challenges	Xie, J., et al.	*IEEE Communications Surveys and Tutorials*	341
7.	2020	A Survey on the Internet of Things (IoT) Forensics: Challenges, Approaches, and Open Issues	Stoyanova, M., et al.	*IEEE Communications Surveys and Tutorials*	301
8.	2020	Thirty Years of Machine Learning: The Road to Pareto-Optimal Wireless Networks	Wang, J., et al.	*IEEE Communications Surveys and Tutorials*	296
9.	2016	An overview of Fog computing and its security issues	Stojmenovic, I., et al.	*Concurrency and Computation: Practice and Experience*	285
10.	2018	The Blockchain as a Decentralized Security Framework [Future Directions]	Puthal, D., et al.	*IEEE Consumer Electronics Magazine*	273
11.	2017	Cyber Threats Facing Autonomous and Connected Vehicles: Future Challenges	Parkinson, S., et al.	*IEEE Transactions on Intelligent Transportation System*	261
12.	2017	Fog of Everything: Energy-Efficient Networked Computing Architectures, Research Challenges, and a Case Study	Baccarelli, E., et al.	*IEEE Access*	249
13.	2017	A Survey on Smart Grid Cyber-Physical System Testbeds	Cintuglu, M.H., et al.	*IEEE Communications Surveys and Tutorials*	247
14.	2017	Fog computing security: a review of current applications and security solutions	Khan, S., et al.	*Journal of Cloud Computing*	240
15.	2017	From Cloud to Fog Computing: A Review and a Conceptual Live VM Migration Framework	Osanaiye, O., et al.	*IEEE Access*	234

**Table 10 sensors-23-08107-t010:** Top-15 highly cited papers in “WOS” DB for “Cybersecurity in Internet of Medical Vehicles”.

S.No.	Publication Year	Publication Title	Authors	Journal Title	Cited By
1.	2017	A Survey on Security and Privacy Issues in Internet-of-Things	Yang, Y.C., et al.	*IEEE Internet of Things Journal*	572
2.	2017	Cyber-Physical Systems Security-A Survey	Humayed, A., et al.	*IEEE Internet of Things Journal*	390
3.	2017	Cost Efficient Resource Management in Fog Computing Supported Medical Cyber-Physical System	Gu, L., et al.	*IEEE Transactions on Emerging Topics in Computing*	240
4.	2019	What are the respiratory effects of e-cigarettes?	Gotts, J.E., et al.	*BMJ-British Medical Journal*	236
5.	2020	An effective feature engineering for DNN using hybrid PCA-GWO for intrusion detection in IoMT architecture	Priya, R.M.S., et al.	*Computer Communications*	172
6.	2018	Internet of Medical Things: A Review of Recent Contributions Dealing with Cyber-Physical Systems in Medicine	Gatouillat, A., et al.	*IEEE Internet of Things Journal*	145
7.	2017	Industry Sponsorship and research outcome	Lundh, A., et al.	*Cochrane Database of Systematic Reviews*	127
8.	2018	PrivacyProtector: Privacy-Protected Patient Data Collection in IoT-Based Healthcare Systems	Luo, E.T., et al.	*IEEE Communications Magazine*	125
9.	2017	A Review of In-Body Biotelemetry Devices: Implantables, Ingestibles, and Injectables	Kiourti, A. and Nikita, K.S.	*IEEE Transactions on Biomedical Transactions on Biomedical Engineering*	120
10.	2016	IDEAL-D: a rational framework for evaluating and regulating the use of medical devices	Sedrakyan, A., et al.	*BMJ-British Medical Journal*	120
11.	2020	Intelligence in the Internet of Medical Things era: A systematic review of current and future trends	Al-Turjman, F., et al.	*Computer Communications*	101
12.	2016	A Triple-Loop Inductive Power Transmission System for Biomedical Application	Lee, B., et al.	*IEEE Transactions on Biomedical Circuits & Systems*	85
13.	2016	A survey on actuators-driven surgical robots	Le, HM., et al.	*Sensors And Actuators a Physical*	84
14.	2016	Security Tradeoffs in Cyber Physical Systems: A Case Study Survey on Implantable Medical Devices	Altawy, H and Youssef, AM	*IEEE Access*	71
15.	2019	Security and Privacy for the Internet of Medical Things Enabled Healthcare Systems: A Survey	Sun, YN, et al.	*IEEE Access*	69

**Table 11 sensors-23-08107-t011:** Typical types of potential cyber-attacks targeting each layer of IOMV applications.

IOMV Layer	Types of Potential Cyber-Attacks at Each IOMV Layer
Network and Servers	• Man in the middle • Session Hijacking • IP Spoofing • Replay • Eavesdropping • Man in the Browser • Buffer Overflow • Mobileware • Ransomware • DoS—denial of service • DDoS—Distributed Denial of Service
IVN	• TCP SYN flood attack • Teardrop attack • Smurf attack • Ping of death attack • Spoofing • ARP—Spoofing • DNS Tunnelling • DNS Hijacking • DNS Spoofing/Poisoning • Cross-Site Scripting Attack • URL Manipulation • Birthday Attack
Network devices	• Phishing
Physical	• Physical Tampering • Malicious code introduction through HMI devices like USBs
IOMV devices	• Encryption Attacks • Privilege Escalation • Brute-Force Password Attack • Credential Stuffing • Password Spraying • Dictionary Attack • Malvertising • DDoS Botnets • Cryptojacking
Software/firmware	• Firmware Hijacking • Botnets • Malware • Code Injection Attacks • Cross-Site Scripting (XSS) • Malvertising • SQL Injection • Malicious Node Injection • Drive-By Attacks (like legit software; fake software; fake warning messages and update prompts; links; spammy websites; drive-by download attacks) • Spyware Attacks • Zero-day exploits
Common attacks irrespective of IOMV layers	• Identity-Based • IOT-Based Attacks • Supply Chain Attacks • Insider Threats • Social Engineering • Illustrative cyber-attacks—(like Mirai Botnet; Verkada Hack (Network and Device); Cold in Finland (DDoS); Jeep Hack (Firmware); Stuxnet (IOT attack))

**Table 12 sensors-23-08107-t012:** Risk Assessment Matrix for qualitative analysis of cybersecurity risk.

	**Severity** 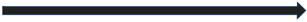
**Likelihood** 		**Severity**	**Low** **(1)**	**Medium** **(2)**	**High** **(3)**	**Critical** **(4)**
**Likelihood**	
**Impossible** **(1)** **(Risk Unlikely to Occur)**	* x 11 * * Low *	* x 12 * * Medium *	* x 13 * * Medium *	* x 14 * * High *
**Possible** **(2)** **(Risk Might Occur)**	* x 21 * * Low *	* x 22 * * Medium *	* x 23 * * High *	* x 24 * * Critical *
**Probably** **(3)** **(Risk Will Occur)**	* x 31 * * Medium *	* x 32 * * High *	* x 33 * * High *	* x 34 * * Critical *

Note: (1) The text represented in the Bold letters are the headings. Colour represents the order of criticality. Green—represents lesser risk, Light Yellow—medium risk, Dark Yellow—Higher risk and Red—Critical risk.

## Data Availability

The data gathered for this research are accessible from the DBs Scopus and WOS.

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
