# Peer review of "Cybersecurity in Internet of Medical Vehicles: State-of-the-Art Analysis, Research Challenges and Future Perspectives"

_sensors, 2023, doi:10.3390/s23198107_

Round 1

Reviewer 1 Report

Dear Authors,

Although there is not special standard or guidance for the cybersecurity of medical vehicles, it is advised to give brief information about  the exist cybersecurity standards and guidance related with medical devices and modern vehicles in the introduction part.

Wish you all success in your researches.

Author Response

==================
Reviewer #1:

Although there is not special standard or guidance for the cybersecurity of medical vehicles, it is advised to give brief information about the exist cybersecurity standards and guidance related with medical devices and modern vehicles in the introduction part.

Wish you all success in your researches.

R1. Provide brief information about existing cybersecurity standards and guidelines related to medical devices and modern vehicles in introduction part.

The Authors Response: The authors are grateful to the reviewer for pointing out to add relevant cybersecurity standards. We have provided the relevant standards, frameworks, and guidelines applicable for medical devices, connected medical devices, and connected vehicles, risk assessments of cybersecurity etc. in the following newly added sections in the revised manuscript as highlighted.

  • Introduction, Section – 1.3 “Related Applicable Standards and Frameworks”
  • and section – 5.3 “Assessment of Cybersecurity Risk”.

Reviewer 2 Report

This analysis is about the upcoming and futuristic trend, Cybersecurity in Internet of Medical Vehicles, which is a niche yet unexplored research-area. And this analysis is done based on the published literature between 2016 and 2023 (Q2) from the databases of Scopus & Web-of-Science. The main objectives of this journal paper regarding this research-field are to evaluate global trends in terms of top publication-outputs, publications-pattern, gaps, and future outlooks etc.., The analysis done in this paper shall help the beginners, relevant industries, and serious researchers to distinguish who are the topmost authors and their contributions to the research work. The analysis was done through various visualisation and free, open-source tools viz. VOSviewer, ScienceScape, GPS-Visualiser, Gephi and WordCloud for pictographic analysis to understand the research-gaps. Based on our analysis, we found that, ‘cybersecurity’ is an extensively used keyword. The paper is organized well. However, there are some points to be considered.

1.  Abstract needs to be more precise highlighting major contributions.

2.  It would be nice to explicitly list the future research directions. Please also discuss some limitations of your proposed method.

3. Please introduce some representative studies of novel approach of artificial general intelligence in the Introduction Section. These researches are critical studies in the field of brain-inspired intelligence to realize high-level intelligence, high accuracy, high robustness, and low power consumption in comparison with the state-of-the-art artificial intelligence works. These researches include: Robust Spike-Based Continual Meta-Learning Improved by Restricted Minimum Error Entropy Criterion; Heterogeneous Ensemble-based Spike-driven Few-shot Online Learning; SAM: A Unified Self-Adaptive Multicompartmental Spiking Neuron Model for Learning with Working Memory; Neuromorphic context-dependent learning framework with fault-tolerant spike routing.

4. The background of the proposed study should be further explained in detail. Some concepts are hard to comprehend without explaining clearly.

6. Grammar is expected to be further improved. Please check the manuscript carefully to remove the typos, improve the language and format.

Grammar is expected to be further improved. Please check the manuscript carefully to remove the typos, improve the language and format.

Author Response

Reviewer #2: 

This analysis is about the upcoming and futuristic trend, “Cybersecurity in Internet of Medical Vehicles”, which is a niche yet unexplored research-area. And this analysis is done based on the published literature between 2016 and 2023 (Q2) from the databases of “Scopus” & “Web-of-Science”. The main objectives of this journal paper regarding this research-field are to evaluate global trends in terms of top publication-outputs, publications-pattern, gaps, and future outlooks etc.., The analysis done in this paper shall help the beginners, relevant industries, and serious researchers to distinguish who are the topmost authors and their contributions to the research work. The analysis was done through various visualisation and free, open-source tools viz. VOSviewer, ScienceScape, GPS-Visualiser, Gephi and WordCloud for pictographic analysis to understand the research-gaps. Based on our analysis, we found that, ‘cybersecurity’ is an extensively used keyword. The paper is organized well. However, there are some points to be considered.

However, few comments and suggestions for further improvement in the manuscript are as follows.

R1.  Abstract needs to be more precise highlighting major contributions.

The Authors Response: The authors are grateful to the reviewer for this critical and valuable comment for pointing out to update the abstract with precisely highlighting the major contributions. We have updated the Abstract precisely and also by highlighting the major contributions. Also, we have reduced the size of the Abstract from approximately 210 words to approximately 183 words. This is modification is providing precise information in further concise format in the revised manuscript.

R2.  It would be nice to explicitly list the future research directions. Please also discuss some limitations of your proposed method.

The Authors Response: The authors are grateful to the reviewer for this critical and valuable comment for pointing out to explicitly listing future directions and limitations of our work. We have added new sections in the revised manuscript like, added Section – 9 Future Outlook and Section – 8 challenges and constraints, this section will help researchers to focus on what challenges and constraints to overcome in future to develop robust cybersecurity measures to counter sophisticated cyberattacks in the IOMV system. Also, we have updated the limitations of our work explicitly in the section – 7 in the revised manuscript as highlighted.

R3. Please introduce some representative studies of novel approach of artificial general intelligence in the Introduction Section. These researches are critical studies in the field of brain-inspired intelligence to realize high-level intelligence, high accuracy, high robustness, and low power consumption in comparison with the state-of-the-art artificial intelligence works. These researches include: Robust Spike-Based Continual Meta-Learning Improved by Restricted Minimum Error Entropy Criterion; Heterogeneous Ensemble-based Spike-driven Few-shot Online Learning; SAM: A Unified Self-Adaptive Multicompartmental Spiking Neuron Model for Learning with Working Memory; Neuromorphic context-dependent learning framework with fault-tolerant spike routing.

The Authors Response: The authors are grateful to the reviewer for this critical and valuable comment for pointing out to add Artificial Intelligence (AIML & AGI) in the introduction. We have added new in the revised manuscript as highlighted, section – 5.2 “AIML techniques for strengthening cybersecurity in IOMVs”. In this section we have discussed backdrop of need of AIML technology in strengthening the cybersecurity in the IOMVs. Various AIML techniques used in existing literature. AIML techniques used for countering cybersecurity issues. Challenges & constraints of AIML techniques. Then need for AGI and EAI techniques including discussing few reviewers indicated research papers, for developing countermeasures to handle increasing sophisticated cyberattacks.

R4 & 5. The background of the proposed study should be further explained in detail. Some concepts are hard to comprehend without explaining clearly.

The Authors Response: The authors are grateful to the reviewer for this critical and valuable comment for pointing out to explain background of the proposed study with detailed explanation. We have updated the introduction section explaining the need of the cybersecurity in the IOMVs and developing countermeasures to handle the cybersecurity risk. Also explicitly mentioned our focus of this paper in the form of Research-Questions in the Introduction section as highlighted. And we tried to explain in detail about each concept in an easy manner wherever is needed and applicable across the entire paper in the revised manuscript.

R6. Grammar is expected to be further improved. Please check the manuscript carefully to remove the typos, improve the language and format.

The Authors Response: The authors are grateful to the reviewer for this critical and valuable comment for pointing out the typo and grammatical errors. We have proofread the whole manuscript and incorporated all the suggestions in the revised manuscript. We have checked the grammatical mistakes and typos using the Grammarly Tool also in the revised manuscript.

Reviewer 3 Report

PFA

Moderate corrections needed.

Author Response

Reviewer #3: 

The authors have done a good job on cybersecurity in the Internet of Medical Vehicles, but this paper still needs improvement.

However, few comments and suggestions for further improvement in the manuscript are as follows.

R1. The abstract is too large, so reduce it to 180 - 200 words if possible.

The Authors Response: The authors are grateful to the reviewer for this critical and valuable comment for pointing out to update the abstract with precisely highlighting the major contributions. We have updated the Abstract precisely and also by highlighting the major contributions. Also, we have reduced the size of the Abstract from approximately 210 words to approximately 183 words. This is modification is providing precise information in further concise format in the revised manuscript.

R2. The difference between the existing survey and the previous survey should be highlighted. Add more of the issues and what is the significance of this analysis.

The Authors Response: The authors are grateful to the reviewer for this critical and valuable comment for pointing out to provide difference between existing and previous surveys and for the reviewer suggestions for adding more issues and explain about the significance of this analysis.

The topic of IOMV and cybersecurity in IOMV is a niche area derived after integrating the IOT applications in connected-vehicles (IOV) and connected-healthcare (IOMT) and our work is one of the pioneer research works in this area. Therefore, there is no straightforward similar surveys on the topic of IOMVs. But we have discussed relevant topics in the corresponding areas of IOT, IOV and IOMT by providing analysis of various existing journals in terms of their focus-areas, novelties, merits and demerits etc. in the Sections-3 & 4, and precisely the Tables – 2, 3, 10 & 11 and Figures-15 & 16.

Also, regarding the topic of adding issues, we are grateful for the reviewer pointing out to adding more issues. We have provided the background of cybersecurity in the IOMVs Section – 1 “Introduction” and specifically added new Section – 1.2. “Risk of Cybersecurity in IOMV Systems. Added new Section – 8 “Challenges & Constraints” for implementing the cybersecurity measures in the IOMV system and Section – 9 “Future Outlook” and added new Section – 5.2 “AIML for strengthening Cybersecurity of IOMVs” for issues regarding implementing countermeasures etc. And we have discussed issues in detailed in the Section – 2 Primary data also.

Also, regarding the topic of significance of this analysis, we are grateful for the reviewer pointing out to highlight significance of this analysis. The analysis done in this paper shall help the beginners, relevant industries, and serious researchers regarding this research topic. For this, we have mentioned and updated the significance of this analysis in the section – 10 “Conclusions” and after Section – 1.3 in 7 bullet points and Section-6 “Results” in the revised manuscript as highlighted.

R3. A proper general IoMT architecture should be included.

The Authors Response: The authors are grateful to the reviewer for this critical and valuable comment for suggesting to include architecture of IOMT. As this topic is related to IOMV, therefore we have added in the revised manuscript as highlighted, a new Section – 1.1 “System of IOMV” in which we have provided the general block diagram of the IOMV architecture and described the system and model of IOMV. Also, discussed in the Section – 1.2 “Risk of Cybersecurity in IOMV System”. With these two newly added sections – 1.1 and 1.2, it will give the understanding of the system model and risk associated with the IOMV system to the researchers.

R4. The use of more parametric comparisons by the authors is recommended. Confusion matrices are a key component for any system’s validity. However, they are seldom mentioned by the authors. Authors should compare all the parameters from existing works.

The Authors Response: The authors are grateful to the reviewer for this critical and valuable comment for suggesting to used parametric comparisons, confusion matrices. As this subject paper of us is only focusing to discuss state-of-the-art analysis regarding the research topic of cybersecurity in the IOMVs, at present in this paper, we are not proposing any method. Therefore, necessity for using confusion matrices is not there, which can be used to evaluate the performance of the classification models. We shall propose new methods and can use confusion matrices in our future research works. But we have added in the revised manuscript as highlighted, the new sections, Section – 5.3 “Assessment of Cybersecurity Risk” and Section – 5.4 “Parameterisation of Cybersecurity”. In these sections we have described the parameters and metrics for how to assess the risk of the cybersecurity in terms of qualitative and quantitative manner, Risk Assessment Matrix, Flow-Chart to address the risk of cybersecurity in the Section – 5.3 and in the section – 5.4, we have described about the parameterisation for assessing the strength of the cybersecurity in the IOMV system etc.

R5. Fig 18 / Fig 21 is unnecessary and doesn’t really contribute to the article. Keyword section is enough.

The Authors Response: The authors are grateful to the reviewer for this critical and valuable comment for excluding the non-contributing figures like Fig – 18 / 21 etc. We have removed such figures and content in the revised manuscript.

R6. Among 93 references, hardly five are from 2023. This shows that the paper hasn’t considered many contemporary related works in the survey. I suggest a few more papers to cite and refer.

  • An artificial intelligence lightweight blockchain security model for security and privacy

in IIoT systems”. J Cloud Comp,(2023), pp.12-38.

  • https://doi.org/10.1155/2022/8457116
  • https://doi.org/10.1007/s11277-021-08725-4
  • https://doi.org/10.1016/j.iot.2023.100852
  • https://doi.org/10.1007/978-3-031-28150-1_6

The Authors Response: The authors are grateful to the reviewer for this critical and valuable comment for pointing out regarding lesser references from 2023. We have taken care including references from 2023 latest by 15-Aug-2023 in this revised manuscript as highlighted.

R7. Unfortunately, the language and sentence structures of this manuscript are at times incomprehensible. The paper needs rewriting and thorough language editing to allow for a proper peer review.

The Authors Response: The authors are grateful to the reviewer for pointing out the language, sentence structure, typo, and grammatical errors. We have proofread the whole manuscript and incorporated all the suggestions in the revised manuscript. We have checked the grammatical mistakes and typos using the Grammarly Tool also.

R8. Authors are advised to follow the IMRAD format for the entire paper.

The Authors Response: The authors are grateful to the reviewer for this critical and valuable comment for the suggestion to revise the manuscript format in the IMRAD structure. We have updated and used similar structure in the revised manuscript. Introduction, Methodology, Analysis & Discussion, Challenges & Constraints, Results, Future-Outlook and conclusion. Also, added new Sections like system of IOMVs, Risk Assessment of Cybersecurity, AIML in Cybersecurity etc. for easy comprehension of the concept and our analysis in the revised manuscript.

Reviewer 4 Report

The authors of this survey "Cybersecurity in Internet of Medical Vehicles: An Analysis" tried to do the analysis last 5/6 years of research done in this field.

Though the overall presentation of the paper is good.

But in this paper, no novelty is there, This is just one study of existing work.

No critical analysis is performed. 

The authors mentioned in Figure 3 Top-10 authors, Figure 4. “Top-10 affiliations, Figure 5. Topmost-10 types of documents, similarly other what is the outcome of this analysis.

There is no technical in-depth analysis like methods/ protocols/algorithms used in this area.

The outcome of this analysis is not clear nor useful to the current research domain.

The contribution of the research is not clearly mentioned.

There is no comparison with existing similar work.

The authors need to improve in all these areas in this paper to be published in quality journal like Sensor. The topic of this research is a good one but the contribution is not up to the mark.

The qualitative or quantitative analysis is not done by the authors.

NA

Author Response

Reviewer #4: 

The authors of this survey "Cybersecurity in Internet of Medical Vehicles: An Analysis" tried to do the analysis last 5/6 years of research done in this field.

Though the overall presentation of the paper is good.

However, few comments and suggestions for further improvement in the manuscript are as follows.

R1. But in this paper, no novelty is there, this is just one study of existing work.

The Authors Response: The authors are grateful to the reviewer for this critical and valuable comment for pointing out that this is just a study of existing work, and no novelty is there. At present the analysis done in this subject paper of us is only focusing to discuss state-of-the-art analysis regarding the research topic of cybersecurity in the IOMVs, and we are not proposing any method. We shall propose new methods in our future research works. But we have added many new sections in the revised manuscript. And the significant contribution of this paper is mentioned after Section – 1.3 in 7 bullet points and Section-6 “Results” as highlighted. These topics are new and important regarding concept of IOMV and risk of cybersecurity in IOMV. Therefore, there is no straightforward similar existing work available in the topic of IOMVs. And our work is one of the pioneer research works in this area.

R2. No critical analysis is performed.

The Authors Response: The authors are grateful to the reviewer for this critical and valuable comment. The topic of IOMV and cybersecurity in IOMV is a niche area derived after integrating the IOT applications in connected-vehicles (IOV) and connected-healthcare (IOMT). And our work is one of the pioneer works on this topic. And we have discussed providing critical analysis of various existing journals in terms of their focus-areas, novelties, merits and demerits etc. in the Sections-3 & 4, and precisely the Tables – 2, 3, 10 & 11 and Figures-15 & 16. Also, we have added new Sections – 8 “Challenges & Constraints” for implementing the cybersecurity measures in the IOMV system. Added new Section – 5.2 “AIML for strengthening Cybersecurity of IOMVs” for critically discussing usage of AIML techniques regarding implementing countermeasures etc. Also, we have mentioned and updated the critical analysis and topics like sections, Section – 5.3 “Assessment of Cybersecurity Risk” and Section – 5.4 “Parameterisation of Cybersecurity”. In these sections we have described the parameters and metrics for how to assess the risk of the cybersecurity in terms of qualitative and quantitative manner, Risk Assessment Matrix, Flow-Chart to address the risk of cybersecurity in the Section – 5.3 and in the section – 5.4, we have described about the parameterisation for assessing the strength of the cybersecurity in the IOMV system, Section – 1.3 “Related Applicable Standards and Frameworks” etc. All these are forming the part of our critical analysis on this research topic in the revised manuscript as highlighted.

R3. The authors mentioned in Figure 3 Top-10 authors, Figure 4. “Top-10 affiliations, Figure 5. Topmost-10 types of documents, similarly other what is the outcome of this analysis.

The Authors Response: The authors are grateful to the reviewer for this critical and valuable comment. The topic of IOMV and cybersecurity in IOMV is a niche area derived after integrating the IOT applications in connected-vehicles (IOV) and connected-healthcare (IOMT) and our work is one of the pioneer research works in this area. Therefore, there is no straightforward similar surveys on the topic of IOMVs. So, this information like Top-10 authors, affiliations etc. are part of the literature analysis and they will provide a critical guidance to the researchers on this topic. However, as suggested we have removed such non-contributing figures and content in the revised manuscript Fig – 18 / 21 etc. (from previous manuscript).

R4. There is no technical in-depth analysis like methods/ protocols/algorithms used in this area.

The Authors Response: The authors are grateful to the reviewer for this critical and valuable comment. As this subject paper of us is only focusing to discuss state-of-the-art analysis regarding the research topic of cybersecurity in the IOMVs, at present in this paper, we are not proposing any methods. We shall propose new methods and can use confusion matrices in our future research works.

But we have added in the revised manuscript as highlighted, the new sections in the revised manuscript which describes in-depth technical details about IOMV Architecture, Relevant Standards & Protocols, Risk Assessment and Parameterisation in Cybersecurity of IOMV systems, Types of Cyberattacks, AIML techniques for developing countermeasures, challenges & constraints, Future-Outlook etc. The following are the newly added sections in the revised manuscript as highlighted.

  • Section – 1.1 “System of IOMV” (Architecture)
  • Section – 1.2 “Risk of Cybersecurity in IOMV System”.
  • Section – 1.3 “Related Applicable Standards and Frameworks”
  • Section – 5.2 “AIML for strengthening Cybersecurity of IOMVs”.
  • Section – 5.3 “Assessment of Cybersecurity Risk”
  • Section – 5.4 “Parameterisation of Cybersecurity”.
  • Added new Section – 8 “Challenges & Constraints”.
  • and Section – 9 “Future Outlook”
  • 1 Types and Characteristics of Cyber-attacks with Table-12: Types of Cyberattacks in IOMV system.

R5. The outcome of this analysis is not clear nor useful to the current research domain.

The Authors Response: The authors are grateful to the reviewer for this critical and valuable comment for pointing out to highlight significance of this analysis. The analysis done in this paper shall help the beginners, relevant industries, and serious researchers regarding this research topic. For this, we have mentioned and updated the significance of this analysis in the revised manuscript in the section – 10 “Conclusions” and after Section – 1.3 in 7 bullet points and Section-6 “Results”, in the revised manuscript as highlighted.

R6. The contribution of the research is not clearly mentioned.

The Authors Response: The authors are grateful to the reviewer for this critical and valuable comment for pointing out about existing similar works.

Also, regarding the topic of adding issues, we are grateful for the reviewer pointing out to adding more issues. We have provided the background of cybersecurity in the IOMVs Section – 1 “Introduction” and specifically added new Section – 1.2. “Risk of Cybersecurity in IOMV Systems. Added new Section – 8 “Challenges & Constraints” for implementing the cybersecurity measures in the IOMV system and Section – 9 “Future Outlook” and added new Section – 5.2 “AIML for strengthening Cybersecurity of IOMVs” for issues regarding implementing countermeasures etc. And we have discussed issues in detailed in the Section – 2 Primary data also.

Also, regarding the topic of significance of this analysis, we are grateful for the reviewer pointing out to highlight significance of this analysis. The analysis done in this paper shall help the beginners, relevant industries, and serious researchers regarding this research topic. For this, we have mentioned and updated the significance of this analysis in the section – 10 “Conclusions” and after Section – 1.3 in 7 bullet points and Section-6 “Results”, in the revised manuscript as highlighted.

R7. There is no comparison with existing similar work.

The Authors Response: The authors are grateful to the reviewer for this critical and valuable comment for pointing out about existing similar works. The topic of IOMV and cybersecurity in IOMV is a niche area derived after integrating the IOT applications in connected-vehicles (IOV) and connected-healthcare (IOMT) and our work is one of the pioneer research works in this area. Therefore, there is no straightforward similar work on the topic of IOMVs. But we have discussed relevant topics in the corresponding areas of IOT, IOV and IOMT by providing analysis of various existing journals in terms of their focus-areas, novelties, merits and demerits etc. in the Sections-3 & 4, and precisely the Tables – 2, 3, 10 & 11 and Figures-15 & 16. Added new Section – 5.2 “AIML for strengthening Cybersecurity of IOMVs” for issues regarding implementing countermeasures etc. in the revised manuscript.

R8. The authors need to improve in all these areas in this paper to be published in quality journal like Sensor. The topic of this research is a good one but the contribution is not up to the mark.

The Authors Response: The authors are grateful to the reviewer for this critical and valuable comment. We have updated the revised manuscript considering all the inputs from the reviewers. The 7 bullet points provided after Section – 1.3 and and Section-6 “Results”, in the revised manuscript as highlighted, explains about the major contributions of this paper. For example, we have added many new sections and updated details like,

  1. I) Newly added Sections:
  • Section – 1.1 “System of IOMV” in which we have provided the general block diagram of the IOMV architecture and described the system and model of IOMV.
  • Section – 1.2 “Risk of Cybersecurity in IOMV System”.
  • Section – 1.3 “Related Applicable Standards and Frameworks”
  • Section – 5.2 “AIML for strengthening Cybersecurity of IOMVs” for issues regarding implementing countermeasures. In this section we have discussed backdrop of need of AIML technology in strengthening the cybersecurity in the IOMVs. Various AIML techniques used in existing literature. AIML techniques used for countering cybersecurity issues. Challenges & constraints of AIML techniques. Then need for AGI and EAI techniques including discussing few reviewers indicated research papers, for developing countermeasures to handle increasing sophisticated cyberattacks.
  • Section – 5.3 “Assessment of Cybersecurity Risk”
  • Section – 5.4 “Parameterisation of Cybersecurity”. In these sections 5.3 and 5.4 we have described the parameters and metrics for how to assess the risk of the cybersecurity in terms of qualitative and quantitative manner, Risk Assessment Matrix, Flow-Chart to address the risk of cybersecurity in the Section – 5.3 and in the section – 5.4, we have described about the parameterisation for assessing the strength of the cybersecurity in the IOMV system etc.
  • 1 Types and Characteristics of Cyber-attacks with Table-12: Types of Cyberattacks in IOMV system.
  • Added new Section – 8 “Challenges & Constraints” for implementing the cybersecurity measures in the IOMV system.
  • and Section – 9 “Future Outlook” etc.

  1. II) Updated details: like we have removed the non-contributing figures and details like Fig – 18 / 21 etc. in the revised manuscript.

R9. The qualitative or quantitative analysis is not done by the authors.

The Authors Response: The authors are grateful to the reviewer for this critical and valuable comment. We have already provided and updated the following details in the revised manuscript as highlighted.

  • We have discussed providing critical analysis of various existing journals in terms of their focus-areas, novelties, merits and demerits etc. in the Sections-3 & 4, and precisely the Tables – 2, 3, 10 & 11 and Figures-15 & 16.
  • We have added new section – 5.2 “AIML techniques for strengthening cybersecurity in IOMVs”. In this section we have discussed backdrop of need of AIML technology in strengthening the cybersecurity in the IOMVs. Various AIML techniques used in existing literature. AIML techniques used for countering cybersecurity issues. Challenges & constraints of AIML techniques. Then need for AGI and EAI techniques including discussing few reviewers indicated research papers, for developing countermeasures to handle increasing sophisticated cyberattacks.
  • Section – 1.1 “System of IOMV” in which we have provided the general block diagram of the IOMV architecture and described the system and model of IOMV.
  • Section – 1.2 “Risk of Cybersecurity in IOMV System”.
  • Section – 1.3 “Related Applicable Standards and Frameworks”
  • Section – 5.2 “AIML for strengthening Cybersecurity of IOMVs” for issues regarding implementing countermeasures. In this section we have discussed backdrop of need of AIML technology in strengthening the cybersecurity in the IOMVs. Various AIML techniques used in existing literature. AIML techniques used for countering cybersecurity issues. Challenges & constraints of AIML techniques. Then need for AGI and EAI techniques including discussing few reviewers indicated research papers, for developing countermeasures to handle increasing sophisticated cyberattacks.
  • Section – 5.3 “Assessment of Cybersecurity Risk”
  • Section – 5.4 “Parameterisation of Cybersecurity”. In these sections 5.3 and 5.4 we have described the parameters and metrics for how to assess the risk of the cybersecurity in terms of qualitative and quantitative manner, Risk Assessment Matrix, Flow-Chart to address the risk of cybersecurity in the Section – 5.3 and in the section – 5.4, we have described about the parameterisation for assessing the strength of the cybersecurity in the IOMV system etc.
  • 1 Types and Characteristics of Cyber-attacks with Table-12: Types of Cyberattacks in IOMV system.
  • Added new Section – 8 “Challenges & Constraints” for implementing the cybersecurity measures in the IOMV system.
  • and Section – 9 “Future Outlook” etc.

All these updates and modifications, we have done in the revised manuscript as highlighted.

Round 2

Reviewer 2 Report

This analysis is about the upcoming and futuristic trend, “Cybersecurity in Internet of Medical Vehicles”, which is a niche yet unexplored research-area. And this analysis is done based on the published literature between 2016 and 2023 from the databases of “Scopus” & “Web-of-Science”. The main objectives of this journal paper regarding this research-field are to evaluate global trends in terms of top publication-outputs, publications-pattern, gaps, and future outlooks etc... However, there is some point to be considered: 1. Neuromorphic computing has been applied to the IOV field. Please discuss this work: Smart traffic navigation system for fault-tolerant edge computing of internet of vehicle in intelligent transportation gateway 2. English should be further improved.

English language should be improved thoroughly.

Reviewer 4 Report

No more comments 

authors did the modification,

Now the paper is ready for acceptance.